# Fractal Patterns in Groundwater Radon Disturbances Prior to the Great 7.9 $M_w$ Wenchuan Earthquake, China

**Aftab Alam [1], Dimitrios Nikolopoulos [2,*] and Nanping Wang [3]**

1 Centre for Earthquake Studies, National Centre for Physics, Shahdra Valley Road, P.O. Box 2141, Islamabad 44000, Pakistan; aftab.alam@ncp.edu.pk
2 Department of Industrial Design and Production Engineering, University of West Attica, Petrou Ralli & Thivon 250, Aigaleo, GR-12244 Athens, Greece
3 Key Laboratory of Geo-Detection, Ministry of Education, School of Geophysics and Information Technology, China University of Geosciences, Beijing 100083, China; npwang@cugb.edu.cn
* Correspondence: dniko@uniwa.gr; Tel.: +30-210-5381338

**Abstract:** This study reports a fractal analysis of one-year radon in groundwater disturbances from five stations in China amidst the catastrophic Wenchuan ($M_w$ = 7.9) earthquake of 12 May 2008 (*day* 133). Five techniques are used (DFA, fractal dimensions with Higuchi, Katz, Sevcik methods, power-law analysis) in segmented portions glided throughout each signal. Noteworthy fractal areas are outlined in the KDS, GS, MSS data, whilst the portions were non-significant for PZHS and SPS. Up to *day* 133, critical epoch DFA-exponents are $1.5 \le \alpha < 2.0$, with several above 1.8. The fractal dimensions exhibit Katz's $D$ around 1.0–1.2, Higuchi's $D$ between 1.5 and 2.0, and Sevcik's $D$ between 1.0 and 1.5. Several power-law exponents are above 1.7, and numerous are above 2.0. All fractal results of the KDS-GS-MSS are further analysed using a novel computerised methodology that locates the exact out-of-threshold fractal areas and combines the outcomes of different methods per five, four, three, and two (maximum 13 combinations) versus nineteen $M_w \ge 5.5$ earthquakes of the greater area. Most coincidences using different techniques are before the great Wenchuan earthquake and after the earthquake. This is not only with one method but with 13 different methods. Other interpretations are also discussed.

**Keywords:** DFA; fractal dimension; Katz; Higuchi; Sevcik; spectral fractal analysis; radon in groundwater; earthquakes; 7.9 $M_w$ Wenchuan earthquake; China

## 1. Introduction

When it comes to severe natural disasters, earthquakes stand out since they may result in great losses of lives and property. Residents of large cities may experience severe effects from the massive quantity of energy produced during an earthquake, especially if the epicentre of the earthquake is near. Catastrophic earthquakes are unavoidable as natural phenomena but are highly challenging to foresee [1]. Consequently, the search for trustworthy seismic precursors is one of the greatest difficulties of science, and significant efforts have been made for many years in this area [2–9]. The issue of earthquake prediction is still open [10]. Earthquakes are inherently complex; thus, several techniques and multi-level strategies are required for prediction [7]. In regions where severe earthquakes are possible, associated predictions call for a gradual downscaling of time, location, and magnitude [2]. Along with the electromagnetic disturbances in ULF, LF, HF, and VHF frequencies, which can indicate approaching earthquakes [4,9], radon-222 (hereafter, radon) is a long-established precursor of impending seismic activity [2,7,11]. Radon is an inert gas created by the radioactive decay of the $^{238}$U series with a half-life of 3.86 days. When it disintegrates, it dissolves in soil pores and liquids before moving on to the surface, subsurface water, and atmosphere. Because radon can travel long distances in water and soil, it can be detected far from its generation location, and due to this, it is very significant

in earthquake-related studies [1]. As the above reviews mention [2,7,11], there is a great number of papers reporting pre-seismic changes in radon in soil gas, groundwater, wells, thermal spas, and atmosphere. As a result, there is a substantial body of studies examining the relationship between the emission of radon and seismic activity [10].

The motivation of the present research is to investigate whether the abnormal behaviour of groundwater radon may be due to the catastrophic Wenchuan ($M_w = 7.9$) shallow (depth = 19 km) earthquake, which occurred on 12 May 2008 (calendar day; hereafter, *day* 133) along the Longmenshan fault (31.0° N, 103.4° E) in Sichuan Province, China [12,13] (Figure 1). This quake was the most destructive one in China since 1976 and the second most devastating seismic shock of this century after the great Sumatra earthquake of 2004 [14]. The groundwater data consist of one-year measurements received from China between 1 January 2008 and 31 December 2008 by five different stations (Figure 1) with epicentral distances between 105.6 km and 526.0 km (Table 1). The possible connection between the variability in geochemical signals and the seismicity due to the Wenchuan earthquake can help to delineate the underlying geophysical processes since the data include information about the subsurface dynamics. Detrended fluctuation analysis, the use of fractal dimensions (FDs) with different methodologies, and power-law fractal analysis are very reliable fractal analysis methods [1,15,16]. These methods can recognise the underlying scaling and long-range features that are characteristic for the unavoidable occurrence of earthquakes even in noisy and non-stationary time series.

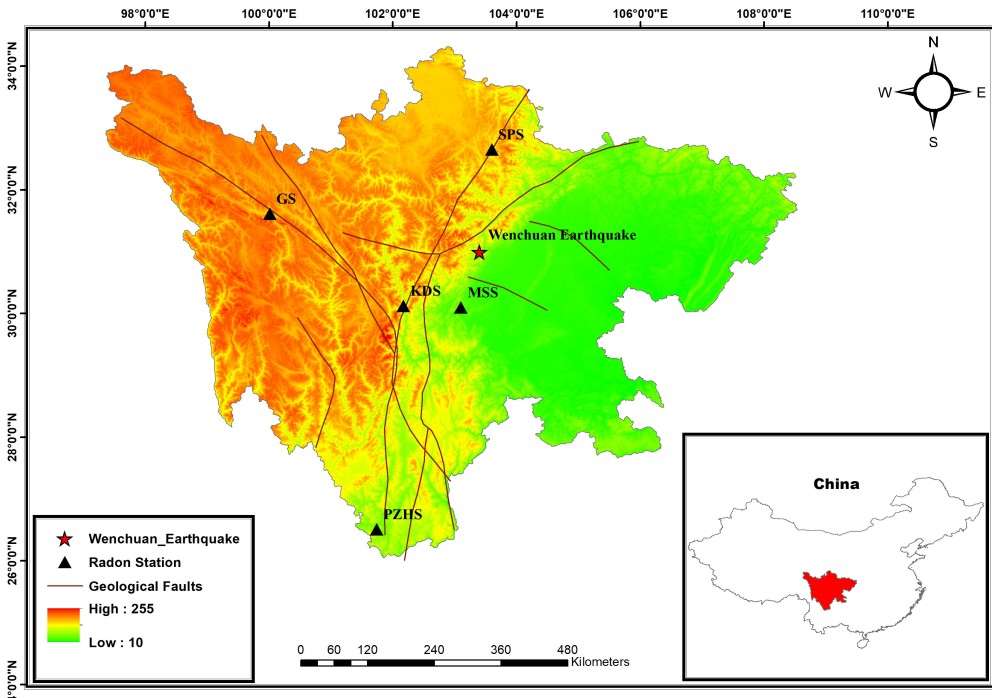

**Figure 1.** Location of the epicentre of the Wenchuan earthquake, showing the radon in groundwater monitoring stations together with elevation data and significant geological faults. The location of the presented area within China is shown in the inserted sub-figure at the bottom right. The codes of stations are given in Table 1. Coloured gradient is the altitude elevation in *m*.

**Table 1.** Radon in groundwater station data. The numbers and coding of the stations are those globally adopted in China, and the names refer to the actual locations. Distance is the epicentral distance.

| Station Code | Station Name | Latitude | Longitude | Distance (km) |
|---|---|---|---|---|
| KDS | Kangding station | 30.12 | 102.17 | 152.2 |
| GS | Ganzi station | 31.61 | 100.01 | 325.5 |
| MSS | Mingshan station | 30.1 | 103.1 | 105.6 |
| PZHS | Panzhihua station | 26.51 | 101.74 | 526.0 |
| SPS | Sonpan station | 32.65 | 103.6 | 182.5 |

## 2. Materials and Methods

### 2.1. Experimental Aspects

2.1.1. Earthquake Activity

The Wenchuan earthquake occurred along the Longmenshan fault (LMSF), which is parallel to eastern Tibet and the Sichuan Basin in the northeast and southwest directions (Figure 1) and is about 500 km long and 40–50 km wide [17]. According to Liu et al. [12], the Wenchuan-Maowen, Yingxiu-Beichuan, and Guanxian-Jiangyou faults, as well as a number of thrust faults produced by the compression of the eastern Tibetan Plateau and the Yangtze Craton, dominate the LMSF zone. Only one significant seismic event ($M_w$ = 6.1) occurred in the LMSF zone before the Wenchuan earthquake, and that was in 1989 [17]. The LMSF zone was dormant until 2008 [12]. The 290 km long segment of the LMSF that was ruptured by the Wenchuan earthquake propagated independently 270 km to the NE and 20 km in the SW direction. The co-seismic surface rupture was 80 km. According to Liu et al. [12], the Wenchuan earthquake was caused by a shift in the LMSF's dip angle (30°–50° SW to 60°–70° NE) and fault motion (SW thrusting motion to NE strike–slip motion). Within 7 days following the mainshock, a series of aftershocks occurred, increased by 5.3 and 10 times more than the standard and relocated catalogues, respectively, [18]. The Wenchuan earthquake disaster resulted in 69,225 fatalities, 374,640 injuries, 17,939 unaccounted-for persons, and over 5 million displaced people.

It is important to emphasise that earthquakes of this magnitude and intensity trigger very important primary and secondary effects, some of which may contribute to the examination of important warning elements [19,20]. Primary effects, such as the surface fault of the Wenchuan earthquake, are associated with changes in concentrations of He, $H_2$, $CO_2$, $CH_4$, $O_2$, $N_2$, Rn, and Hg in soil gas [21]; total electron content anomalies [22]; and well water levels [17]. Primary and secondary effects comprise the need for the introduction of an alternative ESI-07 intensity scale [19,23] and rupture imaging using teleseismic P waves [14].

2.1.2. Measurement Setup

Over the past 50 years, China has developed and established a structured network to track radon levels in groundwater for seismic studies. Almost all of China's provinces are represented in this network, which is made up of stations with a variety of instruments. The network is operated, maintained, and expanded with financial support from the China Earthquake Administration. The supplied radon series may be hourly (high sampling rate—HSR) or daily (low sampling rate—LSR), depending on the sensors deployed at each station. Table 1 lists the LSR stations as Ganzi, Mingshan, Panzhihua, and Sonpan, whereas the only HSR station is Kangding.

To measure the concentration of radon in groundwater, two instruments are used at China's stations. Groundwater is sampled and degassed via bubbling at LSR stations. Then, using a specialised apparatus with the code FD-125, it is directed into an ionisation chamber or ZnS (Ag) detector where the concentration is determined by ionisations or scintillations. For the HSR Kangding station, groundwater is forced via a de-gasser and a gas-collecting device into a ZnS (Ag) detector, where radon is detected by a specialised

instrument coded SD-3A with an hourly sample interval [24]. The accuracy for HSR and LSR measurements is 0.1 Bq/L.

As reported elsewhere [21,25–28], the recorded radon in groundwater concentrations is consistent with the crustal deformative process and/or with the stress diffusion procedure at depths that are able to squeeze deep-seated geofluids towards the surface. Therefore, the related time series include useful information to investigate their potential pre-seismic behaviour.

## 3. Mathematical Aspects

### 3.1. Fractal and Long Memory

Numerous natural physical systems can be explained by fractals. Usual fractal behaviour is often seen when the whole system, or a part of it, is translated, rotated, or stretched in space. Depending on the mathematical description of the changes, the system is characterised either as self-affine or self-similar. Self-affine and self-similar natural systems are fractals in the consensus that any component of them is a little or big copy of the total, but at various scales. As a result, fractal systems may be investigated by targeting their scaled parts. Additionally, the scaling properties of a fractal system are closely associated with the long memory [29–32] and complexity of the system [30,33–36]. Because fractal behaviour, long memory, and complexity are interconnected concepts, examining one of them will usually result in the other. For example, examining a system's complexity will help to determine its long memory and vice versa [1,37,38]. All of these characteristics can show whether a system's past, present, and future are strongly connected.

The direct methods are effective for calculating a system's fractal features [16,37]. The techniques of Katz, Higuchi, and Sevcik are used in this paper because they offer direct estimates for fractal dimension computations [1,39]. Power-law dependencies are also present in fractal systems with long memory and complexity. DFA and power-law analysis are useful tools for the delineation of the related connections [1,37]. These techniques are used in this paper, as well, because of this. By the use of the associated Hurst exponent, all approaches may be compared. These techniques are going to be thoroughly explained in the sections that follow. The Hurst exponent is introduced first, followed by a robust DFA, techniques for the direct calculation of fractal dimensions, and, finally, power-law analysis.

### 3.2. Hurst Exponent

A metric known as the Hurst exponent ($H$) may be used to identify long-lasting connections in both time and space [40,41]. The time evolution of fractal phenomena, as well as the roughness of the related time series, can be identified using the Hurst exponent [1,42,43]. Various research topics have been investigated with the use of Hurst exponent, such as hydrology [40,41], astrophysics and applications [44,45], processes of capital markets [46–49], noisy observations of traces in traffic [50–52], seizures prior to epilepsy [53–55], climatic dynamics [56], and precursory time series before impending earthquakes [1,7,8,57].

The Hurst exponent value offers important details about the time series [1,37,39–41,58–61]:

(i)     If $0.5 < H \leq 1$, then the series has a positive long-range autocorrelation. A series' high value is followed by a series' low value, and vice versa. High Hurst exponents suggest persistent interactions that are predicted to occur in the series' far future;

(ii)    If $0 \leq H < 0.5$, then the low values follow high values in the time series, and vice versa. There is an ongoing exchange between low and high values for low $H$ values in the time series' future (this is known as anti-persistency);

(iii)   If $H = 0.5$ associated processes are random, then the time series are totally uncorrelated.

### 3.3. Detrended Fluctuation Analysis (DFA)

Long-range power-law connections as well as erratic oscillations observed in time series appear prior to earthquakes [1,15,62,63]. DFA is a reliable method for spotting long-

range power-law relationships in noisy, brief, non-stationary signals [1,64]. DFA has been successfully employed in different scientific domains such as the study of changes in the weather and climate [65–68], DNA sequences [69,70], heart dynamics [71–74], urban air pollution [37,39,61], pre-earthquake recordings of radon in soil [25,26], and electromagnetic variations in ULF, kHz, and MHz ranges [1,75–79].

Theoretically, DFA can show if a temporal signal has concealed long-range linkages that result in a self-similar process. Calculating the scaling exponent of the integrated time series allows one to discover these long-term relationships in the initial time series [1,39,57,66,70,71,80–83].

### 3.3.1. Application of DFA

The initial time signal is first integrated. The integrated signal's fluctuations, $F(n)$, are then identified within a window of size $n$. The integrated time series' scaling exponent (self-similarity parameter), $\alpha$, is then calculated by fitting the linear $log(F(n)) - log(n)$ transformation via least squares. The $log(F(n)) - log(n)$ line may display one crossover at a scale $n$, where the slope exhibits an abrupt change, two crossovers at two different scales, $n_1$ and $n_2$ [57], or not even show a crossover at all, depending on the system dynamics.

The DFA of a one-dimensional temporal signal, $y_i$ ($i = 1, \ldots, N$), can be implemented by the following procedure [1,39,57]:

(i)    The initial time series is, first, integrated:

$$y(k) = \sum_{i=1}^{k} (y(i) - \langle y \rangle) \tag{1}$$

The entire average value of the time series is denoted in Equation (1) by the symbol <...>, and $k$ stands for the various time scales.

(ii)    The integrated time series, $y(k)$, is then separated into equal, non-overlapping bins of length $n$.

(iii)    The function $y(k)$ that represents the trend in the bin is then fitted. Simple linear trends or polynomials of second order or higher order may be used. Here, the linear function is used. This linear function's $y$ coordinate is denoted by the notation $y_n(k)$ in each box $n$.

(iv)    The local linear trend, $y_n(k)$, is then subtracted from the integrated time series, $y(k)$, which is detrended in each box of length $n$. The detrended time series, $y_d^n(k)$, is determined in this manner and for each bin as follows:

$$y_d^n(k) = y(k) - y_n(k) \tag{2}$$

(v)    The integrated and detrended time series' fluctuations' root mean square (rms) is then computed for each bin of size $n$ as follows:

$$F(n) = \sqrt{\frac{1}{N} \sum_{k=1}^{N} \left\{ y(k) - y_d^n(k) \right\}^2} \tag{3}$$

where $F(n)$ are the rms fluctuations of the detrended time series, $y_d^n(k)$.

(vi)    For various sizes ($n$) of the scale boxes, the method steps (i)–(v) are repeated. This reveals the specific sort of connection between $F(n)$ and $n$. If there are long-term relationships in the time series, then $F(n)$ and $n$ have an exponential relationship.

$$F(n) \sim n^{\alpha} \tag{4}$$

The DFA scaling exponent $\alpha$ of Equation (4) assesses the strength of the time series' long-term relationships.

(vii)     A linear association between $logF(n)$ and $log(n)$ is found via the logarithmic transformation of Equation (4). A strong linear connection suggests that the accompanying variations are long-lasting and, consequently, have a long memory. The square of Spearman's ($r^2$) is used in this paper to measure the accuracy of the linear fit. Good linear fits are defined as having $r^2 \geq 0.95$ or above [1,15,39,57,84].

### 3.3.2. Sliding Window DFA

The following six-step procedure was followed in order to implement the sliding window DFA:

(a)     The time series was segmented into windows of 64 samples. This segmentation approximately yields a two-month series' part for the PZHS, SPS, GS, and MSS LSR stations, which record one measurement per day. The 64-sample window was also employed for fractal analysis of the data from three monitoring stations of urban air pollution with precisely the same measurement recording rate, namely, one measurement per day [39]. In a recent paper for the PZHS, a 256 segmentation DFA was employed [25], whereas, for radon in soil measurements, an approach of 128-sample window was utilised [85]. Nevertheless, since the windows are shifted 1 sample forward (sliding window technique), the whole signal is analysed, except from a 64-sample window, which was the final one. On the other hand, the 64-sample windowing yields a 64 h window for the HSR station of KDS, i.e., an analysis of about 2.5 days. Despite this differentiation, it is noteworthy that for a radon station in Pakistan, with the same recording rate as the one for KDS, a 64-window analysis was also utilised [16]. DFA from the data of KDS was analysed with 64 sample windows for consistency.

(b)     Every window was fitted using the least-squares fit of $logF(n)$ vs. $log(n)$ in accordance with Equation (4). The data were fitted to a straight line without seeking cross-overs, as in the related literature [1,25,39], with the restriction that the slope of the fit displays a square of Spearman's correlation coefficient above or equal to 0.95.

(c)     The window was advanced by one sample, and the steps (a) and (b) were repeated until the signal's end.

(d)     DFA slopes, $\alpha$, were plotted against time, and the associated plot data were exported to ASCII output files for further use.

### 3.4. Fractal Dimension Analysis

### 3.4.1. Katz's Method

To determine the fractal dimension, $D$, according to Katz's method, the transpose array, $[s_1, s_2, \ldots, s_N]^\top$, of the series, $s_i$, $i = 1, 2, \ldots, N$, is first determined, where $s_i = (t_i, y_i)$ and $y_i$ are the measured series values at the time instances, $t_i$ [86,87].

The value pairs $(t_i, y_i)$ and $(t_{i+1}, y_{i+1})$ correspond to the two following points of the time series ($s_i$ and $s_{i+1}$), for which the Euclidean distance is:

$$dist(s_i, s_{i+1}) = \sqrt{\left(t_i^2 - t_{i+1}^2\right) + \left(y_i^2 - y_{i+1}^2\right)} \tag{5}$$

The distances in Equation (7) add up in a curve, the total length of which is as follows:

$$L = \sum_{i=1}^{i=N} dist(s_i, s_{i+1}) \tag{6}$$

This curve will stretch in the planar to $d$, if it does not cross itself, where $d$ is represented as follows:

$$d = max(dist(s_i, s_{i+1})), i = 2, 3, \ldots, N \tag{7}$$

By combining Equations (5)–(7), the Katz fractal dimension, $D$, becomes the following:

$$D = \frac{log(n)}{log(n) + log(d/L)} \tag{8}$$

where $n = L/\bar{a}$ and $\bar{a}$ is the average value of the distances of the points.

### 3.4.2. Higuchi's Method

To calculate the fractal dimension, $D$, of a time series

$$y(1), y(2), y(3), \ldots, y(N) \tag{9}$$

recorded at intervals $i = 1, 2 \ldots N$, a new sequence, $y_m^k$, is constructed as follows [88–90]:

$$y_m^k : y(m), y(m+k), y(m+2k), \ldots, y(m + \left[\frac{N-m}{k}\right]k) \tag{10}$$

The length of the curve associated with the time series is given by [88]:

$$L_m(k) = \frac{1}{k} \left( \sum_{i=1}^{\left[\frac{N-m}{k}\right]} y(m+ik) - y(m+(i-1)k) \right) \left( \frac{N-1}{\left[\frac{N-m}{k}\right]k} \right) \tag{11}$$

In both equations, $m$ and $k$ are integers that specify the time interval between the series' samples and are connected by the formula $m = 1, 2 \ldots k$, where $[\ldots]$ is the Gauss notation, namely, the bigger integer part of the included value.

By inserting the normalisation factor

$$\frac{N-1}{\left[\frac{N-m}{k}\right]k} \tag{12}$$

the average value, $\langle L(k) \rangle$, of the lengths of Equation (13) exhibits a power law of the following form:

$$\langle L(k) \rangle \propto k^{-D} \tag{13}$$

The Higuchi' s fractal dimension, $D$, is finally calculated by the slope of the linear regression of the logarithmic transformation of $\langle L(k) \rangle$ versus $k$, where $k = 1, 2, \ldots, k_{max}$. It must be noted that the time intervals are $k = 1, .., k_{max}$ for $k_{max} \leq 4$, i.e., $k = 1, 2, 3, 4$ for $k_{max} = 4$ and $k = \left[2^{(j-1)/4}\right]$, where $j = 11, 12, 13 \ldots$, for $k > 4$ ($k_{max} > 4$). Again, $[\ldots]$ is the Gauss notation [87].

### 3.4.3. Sevcik's Method

The fractal dimension of a time series according to the method of Sevcik [91] is approximated from the Hausdorff dimension, $D_h$, as follows [87]:

$$D_h = \lim_{\epsilon \to 0} \left[ -\frac{log(N(\epsilon))}{log(\epsilon)} \right] \tag{14}$$

where $N(\epsilon)$ is the number of segments of length $\epsilon$ that add up to a curve that is associated with the time series. If the length of the curve is $L$, then $N(\epsilon) = L/2\epsilon$ [87] and $D_h$ is the following:

$$D_h = \lim_{\epsilon \to 0} \left[ -\frac{log(L) - log(2\epsilon)}{log(\epsilon)} \right] \tag{15}$$

By applying a linear transformation twice, the *N* points of the curve, *L*, can correspond to a unit square of $N \times N$ cells of the normalised metric space. With this transformation, Equation (15) provides the Sevcik's fractal dimension [87,91]:

$$D_h = \lim_{N \to \infty} \left[ 1 + \frac{log(L) - log(2\epsilon)}{log(2(N-1))} \right] \qquad (16)$$

The calculation improves as $N \to \infty$.

### 3.4.4. Computational Methodology of Fractal Dimension

The next methodology was followed to calculate the fractal dimensions :

(i)　　　As in Section 3.3.2, the time series was segmented into windows of 64 samples. As mentioned, this segmentation approximately corresponds to, for the LSR stations (PZHS, SPS, GS, and MSS), a two-month signal. The 64-sample windowing was also employed in the fractal dimension calculation (with the same methods) from the data of the three LSR urban pollution stations with identical rates of measurements, i.e., one measurement per day [39]. As also mentioned in Section 3.3.2, for the HSR station KDS, the 64-sample segmentation corresponds to approximately 2.5 days. In a previous fractal dimension analysis (with the same methods), a 256-window approach was implemented for the PZHS [25]; however, in a very recent fractal dimension analysis with an identical methodology for an HSR radon station in Pakistan with the same rate of measurements as the one for KDS (one measurement per hour), a 64-window approach was utilised [16]. Finally, as in Section 3.3.2 and for consistency with the windowing of the other stations, a 64-sample window was chosen here as well for the KDS station.

(ii)　　　The fractal dimensions of each method were calculated as follows:

- For the Katz's method: The fractal dimension is the *D* of Equation (8) for $n = 64$ and $\bar{a} = 1$ collected sample per measurement interval (1 day for PZHS, SPS, GS, and MSS and 1 h for the KDS) since $\bar{a}$ corresponds to the distance between the points of the series that constitute the parameter *L* [1,16,39].
- Higuchi's method: Equal to the slope, *D*, of the first-order least-squares fit of the logarithmic transformation of Equation (13), namely, the relation of $log(\langle L(k) \rangle)$ versus $log(k)$, for $kmax = 16$. In the aforementioned analysis for the urban air pollution stations [39], $kmax = 4$, whereas in the analysis of radon in Pakistan [16] and of the electromagnetic disturbances of the Ileia station, Greece, the approach $kmax = 16$ was used. Based on the two latter papers, $kmax = 16$ was also selected here.
- Sevcik's method: Equal to the Hausdorff dimension of Equation (16) ($D = D_h$) for $N = 64$, namely, equal to the number of samples in each window, which constitutes parameter *L*.

(iii)　　Each window was forwarded one sample (sliding window technique), and procedures (i)–(ii) were iterated until the end of the time series.

(iv)　　Time evolution plots of the fractal dimensions in accordance with the Katz's, Higuchi's, and Sevcik's methods were generated, and the partial data were extracted to ASCII files for further use.

### 3.5. Power-Law Analysis

The fractal power-law approach is another robust technique to identify the hidden long-lasting trends that are connected with the long-term links between space and time and are detectable before earthquakes [1,15,57,76,84,92–96]. As with all fractal-based techniques, power-law analysis describes the main core of fractality, namely, the existence of a power-law. Another reason is that the earthquake-generating systems progress gradually to self-organised critical (SOC) states exhibiting fractal evolution in space and time [95]. There have been approaches of fractal power-law analysis based on the Fourier transform [95,96].

The advantageous use of wavelets instead of the Fourier transform has been pointed out in several publications (e.g., [75,97–99]).

A time series' power spectral density, $S(f)$, will follow a power law, if the series is a temporal fractal.

$$S(f) = a \cdot f^{-\beta} \qquad (17)$$

In Equation (17), $\beta$ is the power-law exponent that quantifies the strength of the power-law connection, $a$ is the amplification of the spectral density, and $f$ is the frequency of a transform. According to several publications, this transform was chosen to be the wavelet one based on the Morlet bases and, specifically, $f$ to be the central frequency of the Morlet wavelet [1,15,75,84,92,97–99].

The logarithmic transform of Equation (17) gives the following:

$$\log S(f) = \log a + \beta \cdot \log f \qquad (18)$$

Since Equation (18) is a straight line, the values of $\beta$ and $a$ can be determined by fitting the corresponding data with the least-squares method. As with Section 3.3.2, the goodness of fit of the least-squares fit is quantified by the square of Spearman's ($r^2$) coefficient under the constraint $r^2 \geq 0.95$. The technique has been also described in other publications as spectral fractal analysis or spectral power-law fractal analysis. Hereafter, the phrase "power-law analysis" will be used.

Computational Methodology of Power-Law Analysis

Following the logic of Sections 3.3.2 and 3.4.4, the next steps were followed to implement the power-law analysis:

(a) The time series was separated into 128-sample windows. This is a double window size in comparison to the other two methods. This is performed because power-law analysis does not work well with small-sized windows. For the LSR stations (PZHS, SPS, GS, and MSS), this segmentation corresponds to a 4-month signal and, roughly, a 5-day signal for the KDS. In previous publications, a 128-sample window was employed in $\beta$ parameter estimations [85] for recordings of similar recording rates, whilst in others, a 512-sample window [99] with a recording rate of one measurement every 10 min was employed.

(b) The power spectrum, $S(f)$, based on the Morlet wavelet, as well as the central Morlet frequency, $f$, were calculated in each segment.

(c) The parameters $\log S(f)$ and $\log f$ were fitted via least squares. Exponents, $\beta$, and power amplification, $\alpha$, were computed for every window under the constraint that Spearman's $r^2 \geq 0.95$.

(d) Steps (a) through (c) were iterated to the end of the time series. At each iteration, the window was shifted one sample forwards. As with the other techniques, the whole time series was covered except the last window.

(e) The $\beta$ and $\log a$ data were tabulated and saved in ASCII format for further use.

*3.6. Further Issues*

3.6.1. Formation of Analysis Classes

Two classes are formed to further organise the analysis results:

(a) Class I: This class comprises windows that are associated with DFA least-squares log–log fits with a Spearman's coefficient of $r^2 \geq 0.95$ and, simultaneously, a scaling exponent between $1 < \alpha < 2$, i.e., modelled by the fBm class [84].
Class I segments:

- With anti-persistency ($1.35 < \alpha < 1.5$)–persistency ($1.5 \leq \alpha < 2$) changes, some are of precursory worth [15,57,84,92,99].
- With persistent behaviour ($1.5 \leq \alpha < 2$), they are characterised by others as footprints of impending seismic activity (e.g., [7,8] and the references therein).

(b)　Class II: This class includes time series segments with DFA's $r^2 < 0.95$ (i.e., they do not adhere to the prominent fBm class) or $0 < \alpha < 1$ (i.e., they adhere to the fractional Gaussian noise (fGn) class).
Significantly, Class II segments:

- They are of low precursory worth and low predictability [1,15,57,84,85,92,99].
- They are complements of Class I segments.

3.6.2. Comparisons of the Fractal Results

As shown in previous papers [1,57,84,85], the best approach to compare the results of all fractal methods is through the Hurst exponent.

For Class I segments, the Hurst exponent ($H$) is calculated as follows (e.g., [1,84] and the references therein):

(1)　From DFA's $\alpha$ exponent as:

$$H = \alpha - 1 \tag{19}$$

(2)　From fractal dimension ($D$) as:

$$H = 2 - D \tag{20}$$

(Berry's equation)

(3)　From power-law $\beta$ as:

$$H = 0.5 \cdot (\beta - 1) \tag{21}$$

It should be emphasised that deviations from the straightforward linear connection of Equations (19)–(21) can be seen in the in situ data [1,15,57,84]. The relationship between the fractal analysis parameters remains linear, possibly of a slightly different form, as indicated in the aforementioned works.

*3.7. Meta-Analysis*

The so-called meta-analysis [1,61] is implemented by combining the outcomes from the ASCII files of all five methods, namely, DFA; Higuchi's, Katz's, and Sevcik's fractal dimensions; and power-law analysis. A two-step process is followed:

(a)　(Step 1): According to user-defined thresholds, each ASCII output results file is computationally scanned for out-of-threshold values. The ASCII files carrying the fractal dimension values are searched for under threshold values, whilst the ASCII files containing DFA's exponents and the power-law *beta* values are searched for over threshold values. New ASCII step 1 files are generated that contain the out-of-threshold values.

(b)　(Step 2): Under the restriction that each segment's first sample date is arbitrarily considered as the date of the whole segment, the step 1 ASCII files are computationally filtered to find areas with common dates. The above computational process results in the full coverage of all dates, except the one of the last window. The whole procedure is iterated over the results of all possible combinations of the following:

- DFA versus fractal analysis or versus at least two fractal dimension calculation techniques (six combinations);
- Fractal analysis versus at least two fractal dimension calculation techniques (four combinations);
- One fractal dimension calculation technique versus the other two (three combinations).

Through the above repetitive process, 13 unique combinations of techniques per 5, 4, 3, and 2 are produced. This is very important because it is practically equivalent to the coupling of different mono-fractal methods, and this fact provides a synthetic view of the results of the fractal techniques, increasing the scientific evidence regarding the

underlying nature of the identified fractal disturbances. This has been pointed out in recent publications [1,15,37,39,61,84].

## 4. Results and Discussion

Figures 2–6 present the variation in the DFA exponent, $\alpha$, over time with respect to the evolution of the associated square of the Spearman's correlation coefficient versus the measured disturbances of radon in groundwater. As can be observed from these figures, the DFA scaling exponent profile is entirely distinct from the one of the time series. This has been noted in earlier works as well [1,15,25,27,37,39,57,84]. The reason relies on the fact that DFA manages to locate effectively hidden forms in time series, even in non-stationary ones [62,64,100]. Numerous $\alpha$ exponents are within the Class I range (Section 3.6.1). This means that the corresponding 64-sample windows are successive fBm ones ($r^2 \geq 0.95$), and this has been acknowledged as a notable sign of pre-seismic activity [15,57,84,85,92,99].

The specific details of each station show interesting information. At first, Figure 2 is a noteworthy case of completely different DFA exponents and signal profiles for the KDS. By inspecting this figure, a first period can be observed starting from window 1 (*day* 1) up to window 2100 (approximately *day* 93). During this period, the DFA scaling exponents, $\alpha$, are enhanced with $1.5 \leq \alpha < 2.0$ ($0.5 \leq H < 1.0$, Equation (19)), whereas, most significantly, several exponents are above 1.8 ($0.8 \leq H < 1.0$, Equation (19)). All of these are successive fBm segments since the corresponding Spearman's coefficient in each window is $r^2 \geq 0.95$. These successive fBm segments exhibit persistence since the Hurst exponents are above 0.5, whilst a number of them show great persistent behaviour ($0.8 \leq H < 1.0$). As mentioned in Section 3.6.1, the references therein, and those in this section, the successive fBm segments, and especially those with great persistent behaviour, correspond to radon in groundwater areas with a high potential of association with seismic activity. In addition, in Figure 3, a very interesting period can be observed between windows 50 and 100 (approximately between *day* 60 and *day* 121), with DFA exponents $1.5 \leq \alpha < 2.0$ ($0.5 \leq H < 1.0$) and several above 1.8 ($0.8 \leq H < 1.0$) for the GS. There is a similar case for the MSS. A corresponding high DFA period ($1.5 \leq \alpha < 2.0$, several exponents higher than 1.8) is between windows 70 (*day* 85) and 115 (*day* 139). What makes these three cases of great significance is that the above periods with high DFA exponents are rather synchronous. They are parallel in time, and there are similar periods in these three stations. Indeed, the period for KDS is from *day* 1 to *day* 93, the period for GS is between *day* 60 and *day* 121, and the period for MSS is from *day* 85 to *day* 115. Importantly, these periods correspond to strong persistency since several Hurst exponents are in a range of $0.8 \leq H < 1.0$. They also correspond to the Class I category, exhibiting strong fBm behaviour. The significance of identifying strong, persistent fBm behaviour has been emphasised in several publications [1,15,39,84,97,99] as a footprint for ensuing earthquakes. The interpretations of these precursory footprints in these publications are based on an asperity model [101], according to which it is the roughness of fBm profiles, their relative motion, and their association with micro-crack branching and acceleration that explain why especially persistent fBm behaviour is a noteworthy sign of precursory activity during the preparation of earthquakes. These findings are very important and have to be emphasised, especially because the DFA results of the other two stations (PZHS and SPS) have almost all areas with low DFA exponents, mostly in the Class II category or at the lowest part of the Class I category. The latter fact shows, very clearly, that the PZHS and SPS disturbances are of low precursory value, if not totally insignificant. From the above DFA evidence, it can be highlighted, from a time perspective, that the important time periods identified (KDS from *day* 1 to *day* 93, GS from *day* 60 to *day* 121, and MSS frrom *day* 97 to *day* 130) comprise pre-earthquake signs of the great Wenchuan earthquake, which occurred on *day* 133. According to the literature (see reviews [2,8]), the KDS-GS-MSS time window is within the precursory window range of radon precursors. Particularly, when the high magnitude ($M_w = 7.9$) is accounted for, the above time window extends well before the earthquake's occurrence (even from *day* 1).

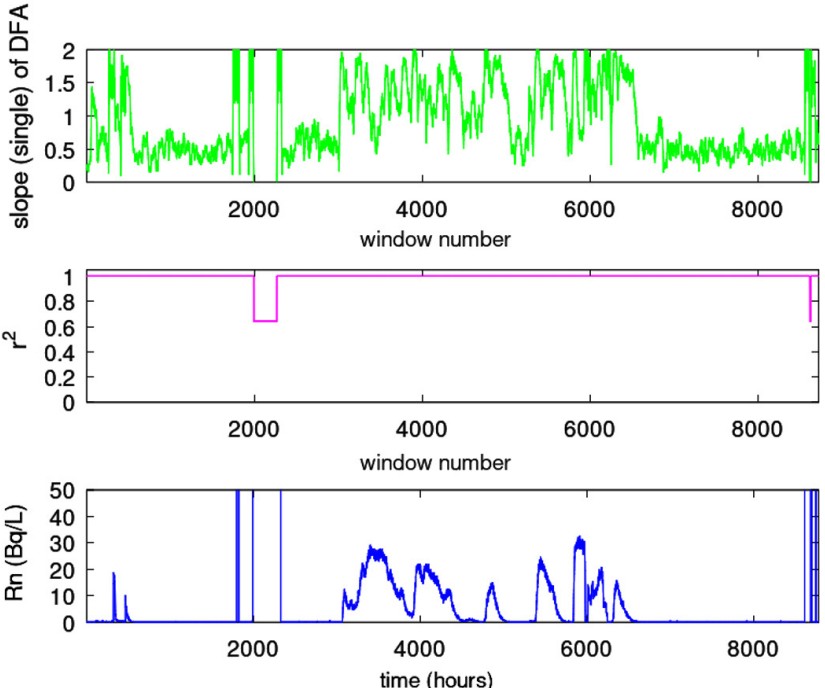

**Figure 2.** Results of DFA. KDS (ID = 3). Window of 64 samples and step of 1 sample. From bottom to top: (**bottom**) radon in groundwater time series; (**middle**) Spearman's correlation coefficient of the goodness of the linear fit of $F(n)$ versus $n$ in every 64-sample window; (**top**) the scaling exponent, $\alpha$ (DFA slope). The horizontal axis is from the beginning (1 January 2008) to the end (31 December 2008) of measurements. The measurement sampling rate is $1\ h^{-1}$. For the window segmentation, please refer to the text.

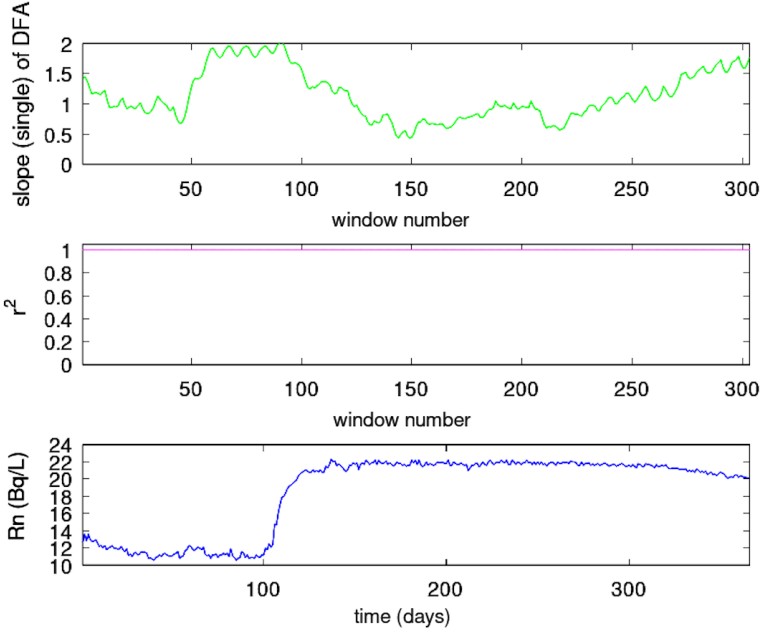

**Figure 3.** Results of DFA. GS (ID = 82). Window of 64 samples and step of 1 sample. From bottom to top: (**bottom**) radon in groundwater time series; (**middle**) Spearman's correlation coefficient of the goodness of the linear fit of $F(n)$ versus $n$ in every 64-sample window; (**top**) the scaling exponent, $\alpha$ (DFA slope). The horizontal axis is from the beginning (1 January 2008) to the end (31 December 2008) of measurements. The measurement sampling rate is $1\ day^{-1}$. For the window segmentation, please refer to the text.

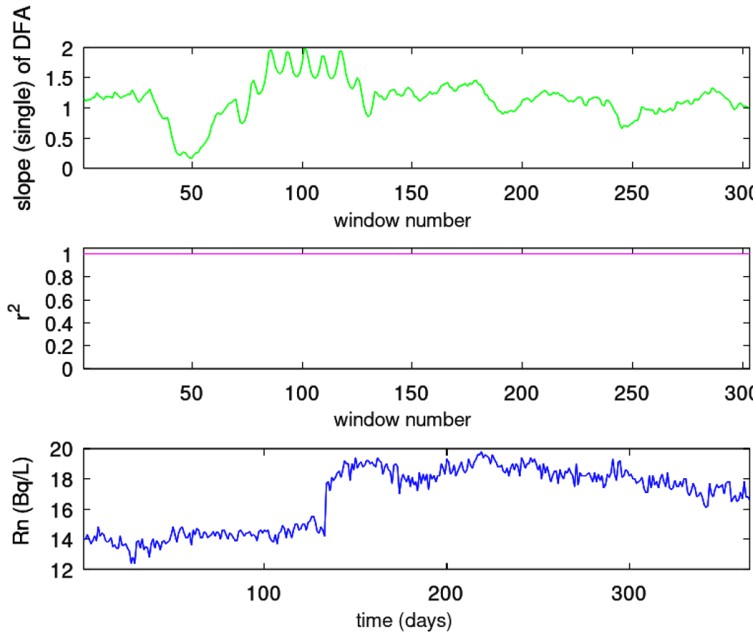

**Figure 4.** Results of DFA. MSS (ID = 83). Window of 64 samples and step of 1 sample. From bottom to top: (**bottom**) radon in groundwater time series; (**middle**) Spearman's correlation coefficient of the goodness of the linear fit of $F(n)$ versus $n$ in every 64-sample window; (**top**) the scaling exponent, $\alpha$ (DFA slope). The horizontal axis is from the beginning (1 January 2008) to the end (31 December 2008) of measurements. The measurement sampling rate is 1 day$^{-1}$. For the window segmentation, please refer to the text.

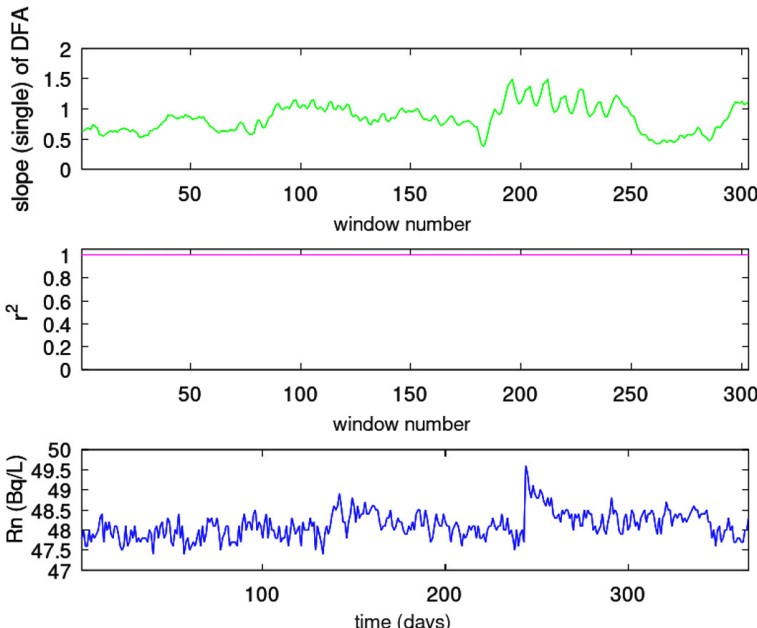

**Figure 5.** Results of DFA. PZHS (ID = 143). Window of 64 samples and step of 1 sample. From bottom to top: (**bottom**) radon in groundwater time series; (**middle**) Spearman's correlation coefficient of the goodness of the linear fit of $F(n)$ versus $n$ in every 64-sample window; (**top**) the scaling exponent, $\alpha$ (DFA slope). The horizontal axis is from the beginning (1 January 2008) to the end (31 December 2008) of measurements. The measurement sampling rate is 1 day$^{-1}$. For the window segmentation, please refer to the  text.

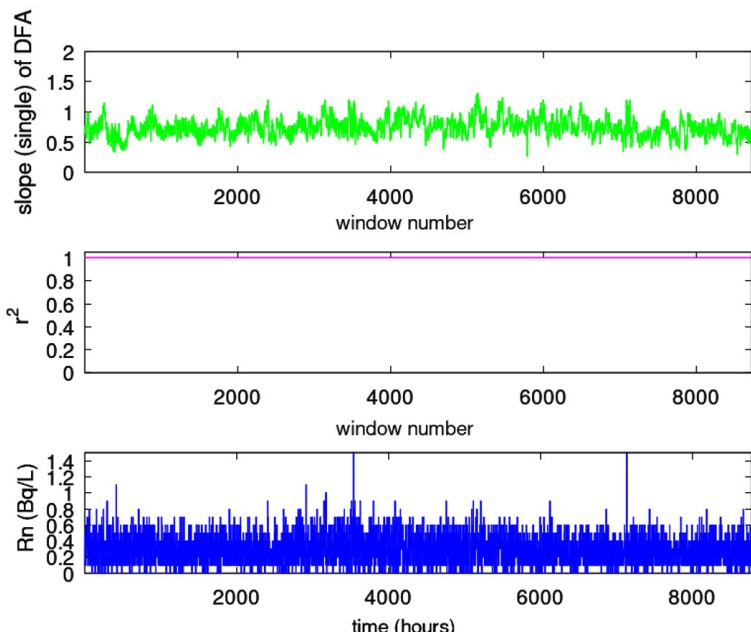

**Figure 6.** Results of DFA. SPS (ID = 149). Window of 64 samples and step of 1 sample. From bottom to top: (**bottom**) radon in groundwater time series; (**middle**) Spearman's correlation coefficient of the goodness of the linear fit of $F(n)$ versus $n$ in every 64-sample window; (**top**) the scaling exponent, $\alpha$ (DFA slope). The horizontal axis is from the beginning (1 January 2008) to the end (31 December 2008) of measurements. The measurement sampling rate is $1\,\mathrm{h}^{-1}$. For the window segmentation, please refer to the text.

The differentiation between the KDS, GS, and MSS and the PZHS and SPS becomes more important when the differentiations in the distance and the underlying geology are taken into consideration. Under the aspect of distance, a potential claim for the PZHS could be that its long epicentral distance (526.0 km) makes it difficult to show significant DFA disturbances. This claim, however, is unsupported when it is considered that the PZHS has shown previous precursory DFA variations for earthquakes with epicentral distances above 533 km [25]. In support, the GS shows precursory DFA variations despite being 325.5 km from the epicentre, especially when the SPS did not show precursory DFA results even though it was located closer (182.5 km) and at a comparable distance with the other two stations (KDS and MSS), which both showed significant DFA outcomes. This is an important observation that has to be outlined. As has been pointed out in the above references [1,15,39,84,97,99], the preparation area of earthquakes includes special preferable precursory paths. Moreover, the selectivity effect that has been proposed and utilised for precursory activity (e.g., [102–105]) suggests that during seismic preparation, there are selective paths that are followed by the disturbed activators. The selection of certain paths has been demonstrated for MHz electromagnetic variations [57,106] and radon precursors [57,107] and has also been expressed in reviews on the subject [2,5,7,8,11]. In this sense, the geological path from a station to the earthquake's epicentre gains special meaning. Hence, it is very important that the GS and KDS are located on the same big fault. More importantly, the MSS is also on the prolongation of this GS-KDS path and is, surely, in line with the Wenchuan's epicentre. This may be supported by the above information that the high elevation of the underlying geology, in association with the big fault on which the GS-KDS-MSS operate and the proximity of the KDS to the Wenchuan epicentre, may explain why these stations are very sensitive to recording radon disturbances with hidden traces of significant precursory values.

This is reinforced by the fact that the SPS-PHZS are on a completely different fault line, despite being on a common fault with the KDS. The fact (as mentioned above) that KDS is also in line with the other two stations (GS and MSS) and near the earthquake's epicentre

makes the recordings of this station even more important. The huge magnitude ($M_w$ = 7.9) and the low depth (19 km) that resulted in a huge energy release, in association with the geologically sensitive background and the proximity of the KDS to the earthquake's epicentre, may explain why post-activity was observed only in this station. Indeed, there is a great period from approximately window 3000 (approximately *day* 133) up to window 6100 (approximately *day* 271) in Figure 2 where the DFA scaling exponents, *α*, are much higher than 1.5, with several above 1.8 ($0.8 \leq H < 1.0$). These DFA variations were identified just after the great Wenchuan earthquake (*day* 133), and this is the first time that such post activity has been found. The reader should note here that there is also an extended period near sample 2000 (approximately *day* 88) and a shorter one near sample 8200 (approximately *day* 363) of completely non-successive ($r^2 < 0.95$) segments. This latter period provided false DFA exponents above 2, and for this reason, they were cut off from the figure. The small period around 8200 is a Class II one. Except for these two non-precursor cases, there are also scattered Class II exponents of low-precursory value. These are non-successive ($r^2 < 0.95$) fBm segments or fGn segments. Finally, with regards to the pre-seismic signs, from a geological perspective, these are in line with the literature findings since, according to the reviews for radon and electromagnetic precursors [2,4–9], the epicentre's distance of precursory activity of KDS-GS-MSS is in accordance with the reported ones. In conclusion, based on the visual observations from the DFA results, the reader should refer to the following points: (a) DFA outlines hidden trends not (usually) observed from the signal; (b) the time window of the identified activity is sufficient to accept it as precursory; (c) the distance and the pathway provide some explanations for the identification in three out of five stations; and (d) most importantly, even for the robust DFA method, visual observation is not enough, and a combination of different methods is necessary in order to find the precursory time periods with enhanced importance and combined evidence [1,37,39,61]. All of these will be presented later in the text, together with the combined evidence.

Figures 7–11 demonstrate the temporal evolution of the fractal dimensions calculated using Katz's, Higuchi's, and Sevcik's methods. There are noticeable differences found. The three methods' computed values for the fractal dimension show variations as well. All discrepancies are due to the varied calculation techniques used by the three fractal dimension techniques. This has been acknowledged in recent publications [1,15,39]. As pointed out, the methods of Katz and Higuchi estimate higher fractal dimensions than the ones of Sevcik. As general trends, the Katz's fractal dimensions are around 1 and 1.1 ($0.9 < H < 1$, Equation (20)), the ones of Higuchi's method are roughly between 1.5 and 2 ($0 < H < 0.5$, Equation (20)), and those of Sevcik's method are, approximately, between 1 and 1.5 ($0.5 < H < 1.0$, Equation (20)). However, what is important is not the value range but the abrupt decrease in the calculated fractal dimensions. For this reason, the details of every sub-figure in Figures 7–11 have great importance, especially in association with the results of the DFA method.

In reference to the KDS and in the first 200 windows (from *day* 1 to *day* 8) of Figure 7, a noteworthy decrease can be observed in Sevcik's fractal dimension between 1 and 1.4 ($0.6 < H < 1.0$) and in Higuchi's *D* between 1 and 2 ($0 < H < 1.0$). Katz's method estimates, unexpectedly, higher fractal dimensions between 1.0 and 1.2($0 < H < 0.8$) in this area. Simultaneously, there is an increase in groundwater radon. This is a rare, serendipitous finding that has been reported in other publications [2,8]. Abrupt drops in Higuchi's fractal dimension are observed, next, between windows 1900 (*day* 84) and 2200 (*day* 97). The drops in Sevcik's fractal dimension are between windows 1800 (*day* 79) and 2000 (*day* 88). The Sevcik's fractal dimensions between windows 2000 and 2200 are above 2, and for this reason, they were cut off, being considered errors in calculations. Once again, the Katz's fractal dimensions are, peculiarly, higher around window 1800. With regards to the GS, a significant decrease in Higuchi's fractal dimension can be spotted in Figure 8 between window 40 (*day* 48) and window 110 (*day* 133). The decrease in Sevcik's fractal dimension is roughly synchronous and of the same profile as the one of Higuchi's,

although milder and with a smaller duration, between windows 60 (*day* 73) and 110 (*day* 133). The fractal dimensions of the MSS (Figure 9) also exhibit decreased profiles through the calculations with Higuchi's and Sevcik's methods. These decreases in *D* values can be observed between windows 70 (*day* 85) and 140 (*day* 170). In summary, here, the key periods from the fractal dimension calculations for the KDS are between *days* 1 and 8 and between *days* 84 and 97. For the GS, the period is between *day* 48 and *day* 110, and for the MSS, the period is between *day* 85 and *day* 170. Since the Wenchuan earthquake occurred on *day* 133, the fractal dimension variations of KDS and GS can be considered, most probably, pre-seismic. The same is valid for the *day* 85–*day* 133 variations of the MSS. These findings reinforce the claims expressed in the discussion of DFA above since the fractal dimension calculation techniques are completely different from the ones of DFA. Moreover, two fractal techniques show these tendencies and, significantly, in comparable time intervals. These facts further support the necessity of using different fractal techniques in parallel. This has been emphasised in recent publications [1,15,61,84]. However, as with the DFA outcomes of the KDS, there is a wide period between windows 3000 (*day* 133) and 6100 (*day* 271) during which the fractal dimensions from Higuchi's and Sevcik's methods exhibit very significant variations. This period is the same as the corresponding one discussed for the DFA results and refers to post-seismic variations. Additional post-seismic variations are addressed here via Higuchi's and Sevcik's fractal dimension from the MSS between *day* 133 and *day* 170. The reader should note here that as with the outcomes of DFA, SPSS, as shown in Figure 11, does not show certain *D* patterns. Despite the fact that DFA did not provide trends in DFA profiles for the PZHS, the corresponding Higuchi's fractal dimension shows a small decrease between windows 200 and 250.

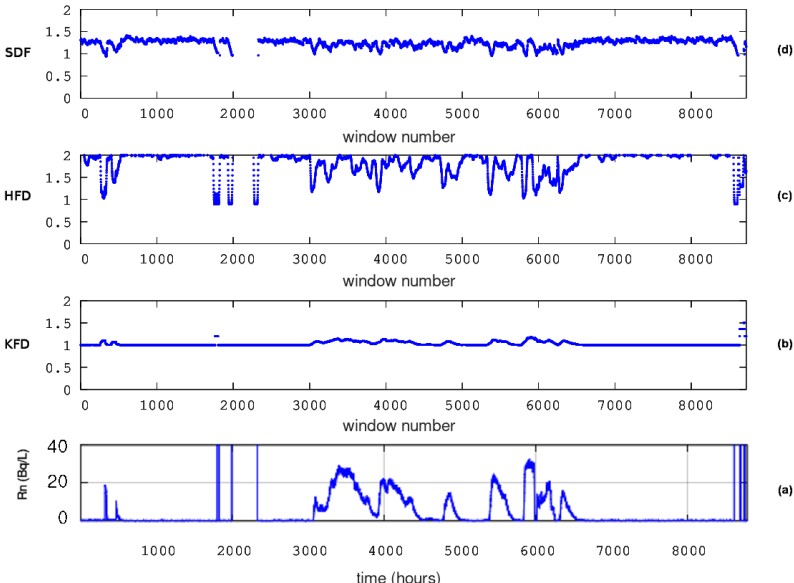

**Figure 7.** Results from fractal dimension analysis. KDS (ID = 3). Window of 64 samples, 16 subcategories of Higuchi's method, and step of 1 sample. From bottom to top: (**a**) the radon in groundwater time series and the fractal dimensions according to the algorithms of (**b**) Katz (KFD), (**c**) Higuchi (HFD), and (**d**) Sevcik (SFD). The horizontal axis is from the beginning (1 January 2008) to the end (31 December 2008) of measurements. The measurement sampling rate is 1 h$^{-1}$. For the window segmentation, please refer to the text.

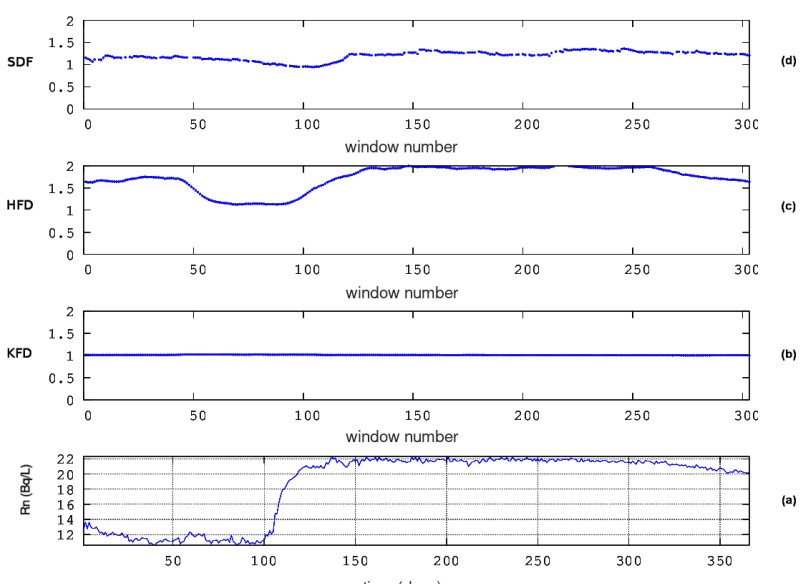

**Figure 8.** Results from fractal dimension analysis. GS (ID = 82). Window of 64 samples, 16 subcategories of Higuchi's method, and step of 1 sample. From bottom to top: (**a**) the radon in groundwater time series and the fractal dimensions according to the algorithms of (**b**) Katz (KFD), (**c**) Higuchi (HFD), and (**d**) Sevcik (SFD). The horizontal axis is from the beginning (1 January 2008) to the end (31 December 2008) of measurements. The measurement sampling rate is 1 day$^{-1}$. For the window segmentation, please refer to the text.

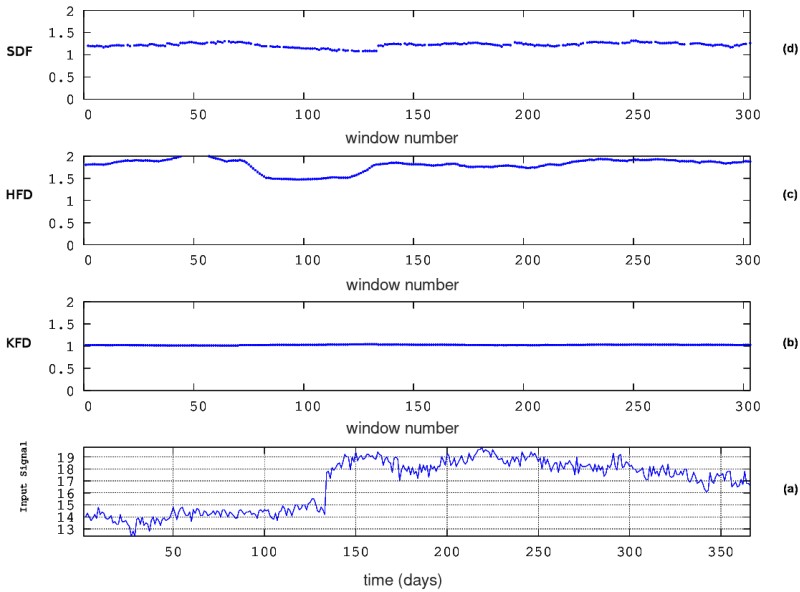

**Figure 9.** Results from fractal dimension analysis. MSS (ID = 83). Window of 64 samples, 16 subcategories of Higuchi's method, and step of 1 sample. From bottom to top: (**a**) the radon in groundwater time series and the fractal dimensions according to the algorithms of (**b**) Katz (KFD), (**c**) Higuchi (HFD), and (**d**) Sevcik (SFD). The horizontal axis is from the beginning (1 January 2008) to the end (31 December 2008) of measurements. The measurement sampling rate is 1 day$^{-1}$. For the window segmentation, please refer to the text.

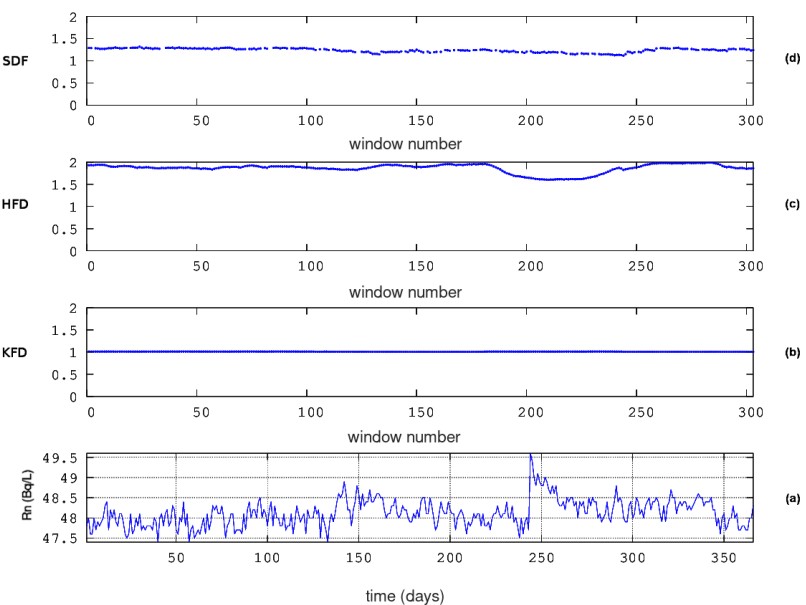

**Figure 10.** Results from fractal dimension analysis. PZHS (ID = 143). Window of 64 samples, 16 sub-categories of Higuchi's method, and step of 1 sample. From bottom to top: (**a**) the radon in groundwater time series and the fractal dimensions according to the algorithms of (**b**) Katz (KFD), (**c**) Higuchi (HFD), and (**d**) Sevcik (SFD). The horizontal axis is from the beginning (1 January 2008) to the end (31 December 2008) of measurements. The measurement sampling rate is $1\ \text{day}^{-1}$. For the window segmentation, please refer to the text.

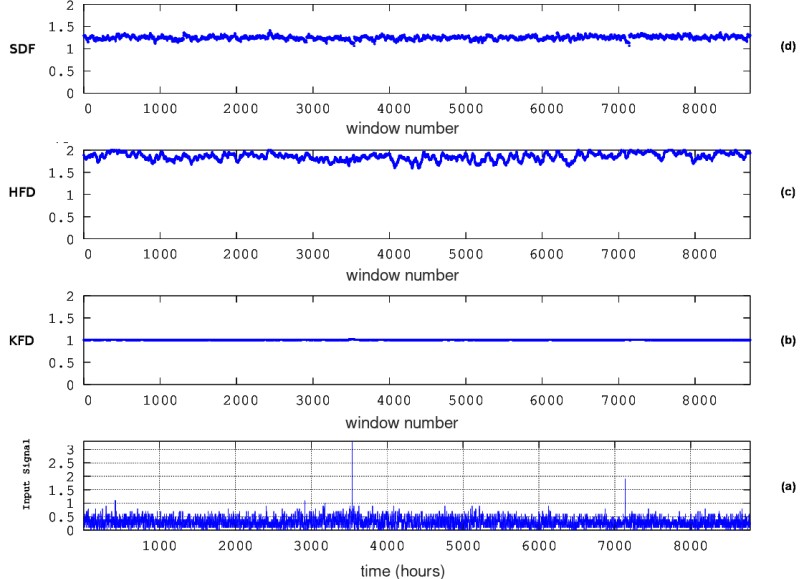

**Figure 11.** Results from fractal dimension analysis. SPS (ID = 149). Window of 64 samples, 16 sub-categories of Higuchi's method, and step of 1 sample. From bottom to top: (**a**) the radon in groundwater time series and the fractal dimensions according to the algorithms of (**b**) Katz (KFD), (**c**) Higuchi (HFD), and (**d**) Sevcik (SFD). The horizontal axis is from the beginning (1 January 2008) to the end (31 December 2008) of measurements. The measurement sampling rate is $1\ \text{h}^{-1}$. For the window segmentation, please refer to the text.

Figures 12–16 show the results from the power-law method. This method is one of the wider used techniques and has been considered as one of the most powerful to identify

the hidden patterns in time series [43,75–77,92,95,98,99,106,108–115]. Hence, when certain trends are found with the power-law method, there is strong evidence for the underlying long memory of the associated geosystem. At first, as with the outcomes of the DFA and fractal dimension methods, the time evolution of the power-law exponent, $\beta$, differs from the one of the time series. However, in order to discuss the interesting results of each figure, the following information should be taken into consideration for the successive fractal segments:

1. If $1.0 < \beta \leq 3.0$, then the associated time series is a temporal fractal and follows the Class I category:

    - If $1.0 < \beta < 2.0$, then the time series follows anti-persistent paths;
    - If $2.0 < \beta < 3.0$, then the time series follows persistent paths.

2. If $-1.0 \leq \beta < 1.0$, then the time series is of low predictability and follows the Class II category:

    - If $\beta = 1.0$, then the fluctuations in the related processes are not growing and, hence, a stationary system describes the series;
    - If $\beta = 2.0$, then the underlying dynamics are random and the related system has no memory.

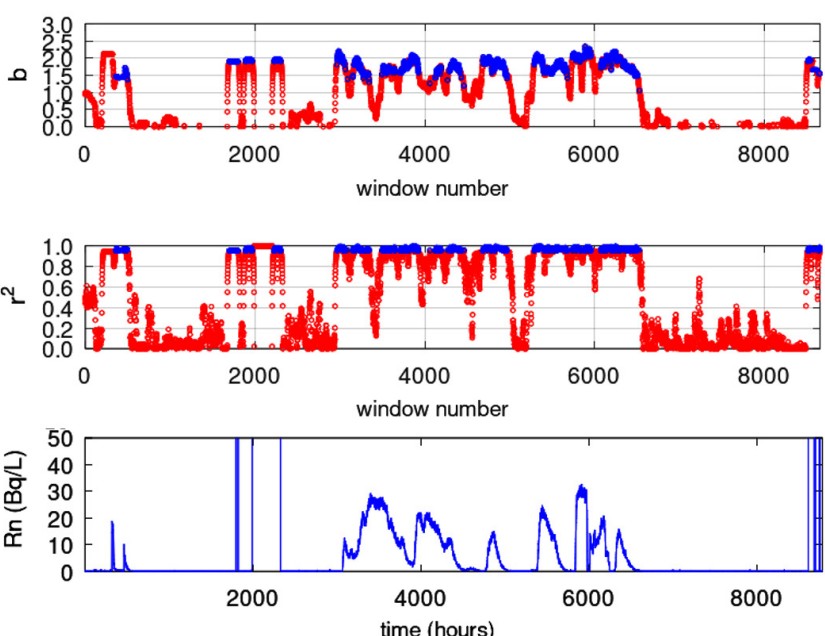

**Figure 12.** Results from fractal analysis: KDS (ID = 3). Window of 128 samples and step of 1 sample. From bottom to top: (**bottom**) radon in groundwater time series; (**middle**) Spearman's correlation coefficient of the goodness of the linear fit of Equation (18); (**top**) time evolution of power-law $\beta$ exponent. Blue areas represent the successive ($r^2 \geq 0.95$) fractal windows. Red areas are non-successive windows. The horizontal axis is from the beginning (1 January 2018) to the end (31 December 2018) of measurements. The measurement sampling rate is $1\,\text{h}^{-1}$. For the window segmentation, please refer to the text.

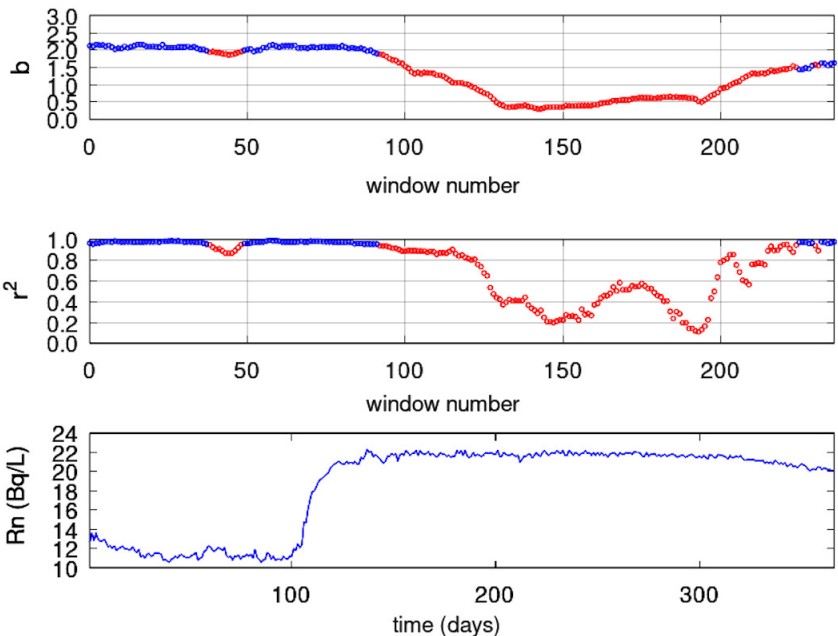

**Figure 13.** Results from fractal analysis: GS (ID = 82). Window of 128 samples and step of 1 sample. From bottom to top: (**bottom**) radon in groundwater time series; (**middle**) Spearman's correlation coefficient of the goodness of the linear fit of Equation (18); (**top**) time evolution of power-law $\beta$ exponent. Blue areas represent the successive ($r^2 \geq 0.95$) fractal windows. Red areas are non-successive windows. The horizontal axis is from the beginning (1 January 2018) to the end (31 December 2018) of measurements. The measurement sampling rate is 1 day$^{-1}$. For the window segmentation, please refer to the text.

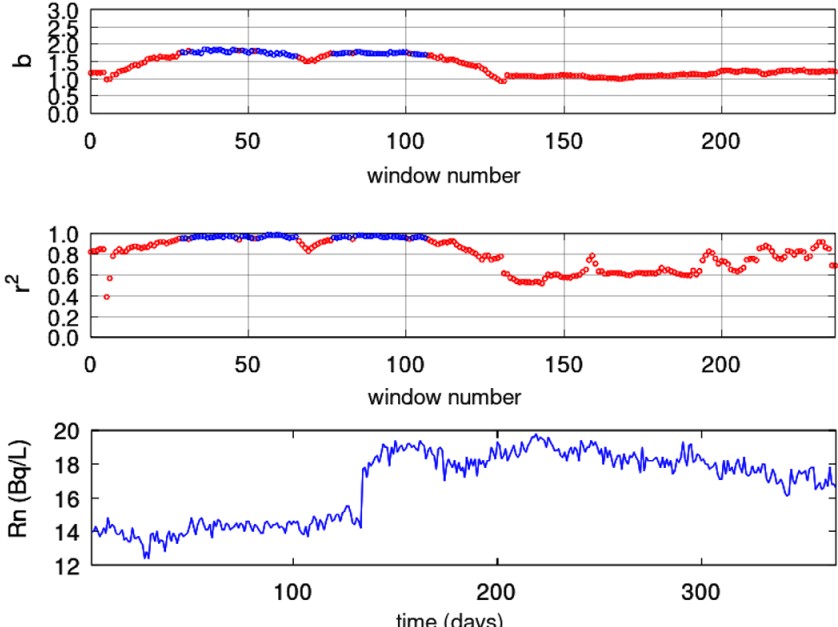

**Figure 14.** Results from fractal analysis: MSS (ID = 83). Window of 128 samples and step of 1 sample. From bottom to top: (**bottom**) Radon in groundwater time series; (**middle**) Spearman's correlation coefficient of the goodness of the linear fit of Equation (18); (**top**) Time evolution of power-law $\beta$ exponent. Blue areas represent the successive ($r^2 \geq 0.95$) fractal windows. Red areas are non-successive windows. The horizontal axis is from the beginning (1 January 2018) to the end (31 December 2018) of measurements. The measurement sampling rate is 1 day$^{-1}$. For the window segmentation, please refer to the text.

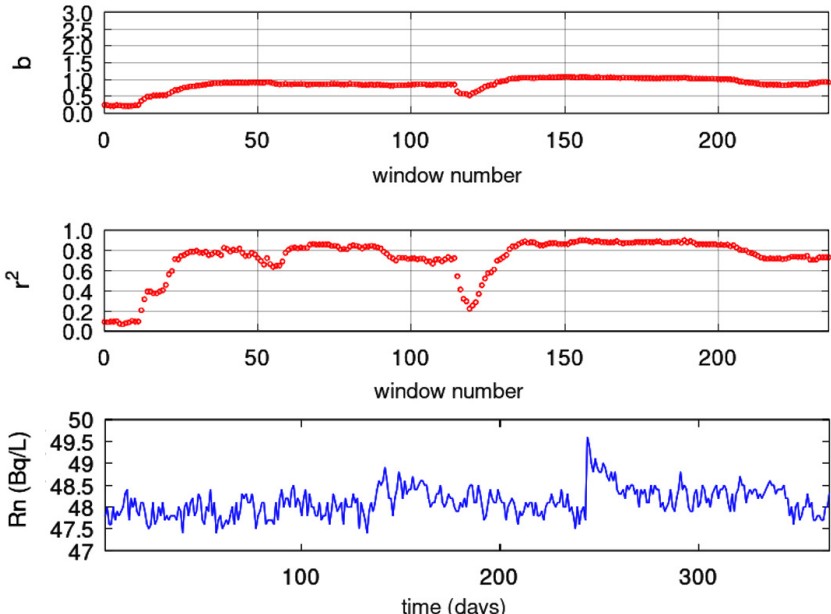

**Figure 15.** Results from fractal analysis: PZHS (ID = 143). Window of 128 samples and step of 1 sample. From bottom to top: (**bottom**) radon in groundwater time series; (**middle**) Spearman's correlation coefficient of the goodness of the linear fit of Equation (18); (**top**) time evolution of power-law $\beta$ exponent. Blue areas represent the successive ($r^2 \geq 0.95$) fractal windows. Red areas are non-successive windows. The horizontal axis is from the beginning (1 January 2018) to the end (31 December 2018) of measurements. The measurement sampling rate is 1 day$^{-1}$. For the window segmentation, please refer to the text.

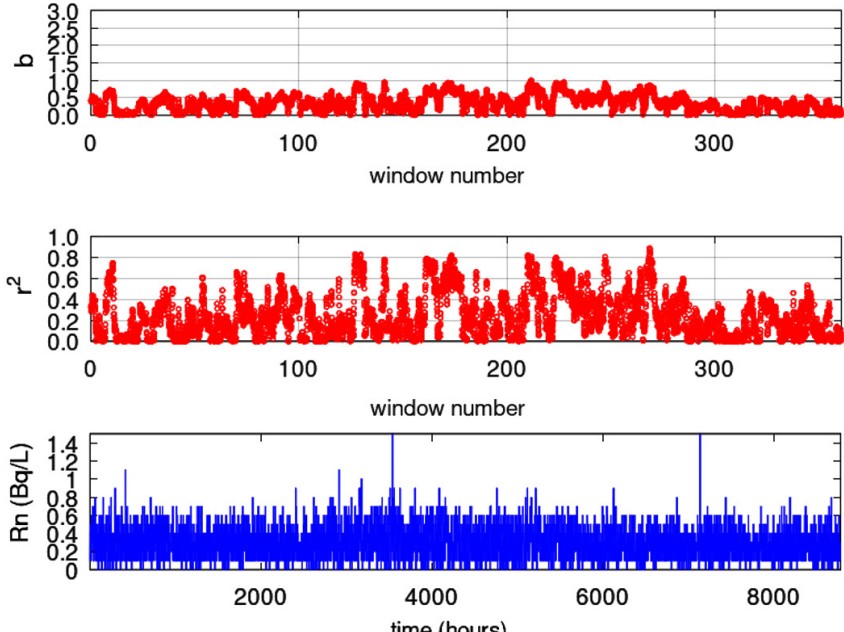

**Figure 16.** Results from fractal analysis: SPS (ID = 149). Window of 128 samples and step of 1 sample. From bottom to top: (**bottom**) radon in groundwater time series; (**middle**) Spearman's correlation coefficient of the goodness of the linear fit of Equation (18); (**top**) time evolution of power-law $\beta$ exponent. Blue areas represent the successive ($r^2 \geq 0.95$) fractal windows. Red areas are non-successive windows. The horizontal axis is from the beginning (1 January 2018) to the end (31 December 2018) of measurements. The measurement sampling rate is 1 h$^{-1}$. For the window segmentation, please refer to the text.

In Figure 12, for the KDS, as with DFA and the fractal dimension calculation techniques, a first period is observed up to window 2200 (approximately *day* 120) with scattered successive fractal windows with $1.7 < b < 2.2$. These fractal epochs correspond to predictable Class I segments with an interchange between persistency and anti-persistency. According to several publications (e.g., [99]), this is a sign of precursory activity. Interestingly, this epoch is almost identical to those identified with DFA and the three fractal dimension techniques, and this is very important. Figure 13 for the GS shows a first period between windows 1 and 40 (*day* 1 to *day* 58) and windows 50 and 90 (*day* 73 to *day* 131). Both these periods have areas with $b > 2$ (Class I, persistent) and can be considered as precursory of the great Wenchuan earthquake. The last period at the end of the analysed windows shows $b < 1.6$ and, as with DFA and fractal dimension techniques, has low predictability and is, hence, of low precursory value. This combined finding provides more evidence. In Figure 14, for the MSS, there are two periods, between window 30 (*day* 43) and window 60 (*day* 87) and between window 70 (*day* 102) and window 110 (*day* 160). Although these periods are anti-persistent, they are within the middle part of the predictable value range of the Class I category. Since the periods match with those of the other techniques, there is a probability that they might be signs (pre and post) of the great Wenchuan earthquake. As with the DFA and fractal dimension techniques, there is a great post-seismic region within the same period. It is very important that the findings of the techniques match even for the two other stations, the PZHS and SPS. Both showed no successive fractal window. This fact reinforces the findings of the DFA and Higuchi's and Sevcik's methods for the MSS.

Evaluating the results presented so far, there is a period up to *day* 133 (the day of occurrence of the great Wenchuan earthquake) that can be systematically identified in all fractal analysis results, that is, from all methods for the KDS, GS, and MSS, whereas the PSZH and SPS do not show any such noteworthy period. These fractal epochs are discussed and considered pre-seismic signatures of the great Wenchuan earthquake. The recognition of signs in the KDS, GS, and MSS and the lack of systematic signs in the PSZH and SPS can be attributed to the different geological paths, in association with the theory of asperities and the selectivity effects in earthquake-related research. It is also extensively discussed that finding periods of over- or under-threshold fractal values is significant, but most significant is the synthetic finding of common over- or under-threshold areas with different techniques (meta-analysis, Section 3.7). When, and if, such common locations are discovered, the scientific arguments for the existence of a seismic warnings concealed in the time series increases, making these claims stronger. It is, therefore, very important to apply meta-analysis to the fractal results of the KDS, GS, and MSS so as to enhance the presented evidence. Meta-analysis for the PSZH and SPS results is unnecessary since the corresponding signs are not enough. On the other hand, an earthquake as great as the Wenchuan is expected to be associated with other earthquakes. Moreover, other earthquakes in the overall area might also explain the warnings, and there is a possibility that the presumed post-seismic signs might also be the pre-seismic activity of other earthquakes in the area, and this is the first time such a view is mentioned in this paper. For this purpose, the data on earthquakes of 2008 with $M_w \geq 5.5$ were accessed from USGS over an area greater than the inserted one in Figure 1 [116]. The latter reference presents online the selected area together with data on 19 earthquakes and a map representing it on Chinese terrain. The data on these 19 earthquakes are presented in Table 2, whereas Figure 17 presents the earthquakes in a Google Earth Map after creating the corresponding map kml file from USGS and importing it to Google Earth. In this way, Figure 17 presents the earthquakes alternatively in comparison to the USGS map [116].

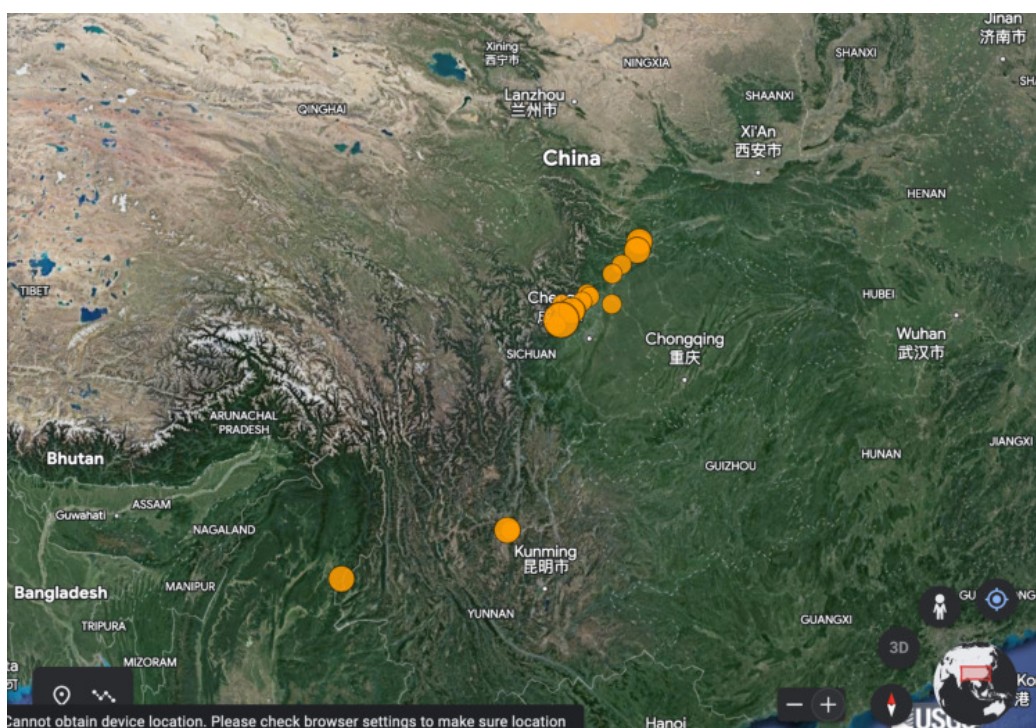

**Figure 17.** Location of the earthquakes of 2008 with $M_w \geq 5.5$ over an area greater than the one inserted in Figure 1. This figure was created with Google Earth using a kml file from USGS. Greater circles are earthquakes with greater magnitude $M_w$. The biggest circle near Sichuan is the great Wenchuan earthquake. The 19 presented earthquakes are shown in Table 2.

**Table 2.** Earthquakes of 2008 with $M_w \geq 5.5$ in China for the area presented in Figure 17. The last event (i/i 19) is the great Wenchuan earthquake

| i/i | Year | Month | Day | Hour | Minute | Second | $M_w$ | Latitude | Longitude | Depth (km) |
|-----|------|-------|-----|------|--------|--------|-------|----------|-----------|------------|
| 1 | 2008 | 8 | 31 | 8 | 31 | 10 | 5.6 | 26.232 | 101.97 | 10 |
| 2 | 2008 | 8 | 30 | 8 | 30 | 53 | 6.0 | 26.241 | 101.889 | 11 |
| 3 | 2008 | 8 | 21 | 12 | 24 | 30 | 6.0 | 25.039 | 97.697 | 10 |
| 4 | 2008 | 8 | 5 | 9 | 49 | 17 | 6.0 | 32.756 | 105.494 | 6 |
| 5 | 2008 | 8 | 1 | 8 | 32 | 43 | 5.7 | 32.033 | 104.722 | 11 |
| 6 | 2008 | 7 | 24 | 9 | 30 | 9 | 5.7 | 32.747 | 105.542 | 10 |
| 7 | 2008 | 7 | 23 | 19 | 54 | 44 | 5.5 | 32.752 | 105.498 | 4 |
| 8 | 2008 | 5 | 27 | 8 | 37 | 51 | 5.7 | 32.71 | 105.54 | 10 |
| 9 | 2008 | 5 | 25 | 8 | 21 | 49 | 6.1 | 32.56 | 105.423 | 18 |
| 10 | 2008 | 5 | 17 | 8 | 25 | 48 | 5.8 | 32.24 | 104.982 | 9 |
| 11 | 2008 | 5 | 16 | 5 | 25 | 47 | 5.6 | 31.355 | 103.351 | 3 |
| 12 | 2008 | 5 | 13 | 7 | 7 | 8 | 5.8 | 30.89 | 103.194 | 9 |
| 13 | 2008 | 5 | 12 | 20 | 8 | 50 | 5.6 | 31.413 | 103.889 | 21.7 |
| 14 | 2008 | 5 | 12 | 11 | 11 | 2 | 6.1 | 31.214 | 103.618 | 10 |
| 15 | 2008 | 5 | 12 | 9 | 42 | 24 | 5.5 | 31.527 | 104.092 | 10 |
| 16 | 2008 | 5 | 12 | 6 | 43 | 14 | 5.8 | 31.211 | 103.715 | 10 |
| 17 | 2008 | 5 | 12 | 6 | 42 | 8 | 5.7 | 31.342 | 104.682 | 10 |
| 18 | 2008 | 5 | 12 | 6 | 61 | 56 | 5.7 | 31.586 | 104.032 | 10 |
| 19 | 2008 | 5 | 12 | 6 | 28 | 1 | 7.9 | 31.002 | 103.322 | 19 |

In the consensus expressed above, the final step of the related analysis should include the following: (a) the above 19 earthquakes (Wenchuan included); (b) the fractal results and the discussion on threshold setting presented in Figures 2–16; and (c) the logic of connecting the fractal methods expressed in Section 3.7. To achieve this, Figure 18 presents combined plots for each station. This multi-figure conceals all the important findings. Similar representation has been adopted in other publications as well [1,25,37,39]. The subplots

are mixed and have several symbols. For this reason it is crucial to delineate the significant information given:

(a)     All over- or under-threshold results of all fractal methods (step 1, meta-analysis) for the KDS, MSS, and GS. The threshold results of each station are combined per 2, 3, 4, and 5 methods (step 2, meta-analysis; a total of 13 combinations) versus all 19 earthquakes of Table  2 and Figure 17.

As mentioned in the previous paragraph, it is important not only to identify foot-prints using one or more techniques (already conducted here) but more importantly to link the different techniques focusing on similar aspects of the problem at hand. To achieve this:

(1)     The exact over- or under-threshold dates were located computationally from the fractal outputs of each station (step 1, meta-analysis). These dates are year, month, day, and hour for the HSR and KDS, and year, month, and day for the MSS and GS. This is conducted through a serial search.

(2)     The common threshold dates from all different techniques were found through an incremental computational search. The outputs used are from all methods and with up to 13 different combinations of these. All outputs were generated through special software and were stored in a computer for use.

(3)     The earthquake data from USGS [116] were transformed into an adequate ASCII file for the generation of the final plot.

(b)     Wherever the symbols of different methods coincide in time, this means that the signs of seismicity are provided by more than one method. If all 13 methods coincide, this means that the evidence is maximised. The more the techniques point to similar findings, the more rigid the evidence is. It should be emphasised to the reader that this coinciding is conducted on the step 1 results, that is, on the fractal outputs.

In the above sense and starting from the combined findings of lesser importance, it can be observed from sub-figure a of Figure 18 that there are several windows of the combination of DFA versus Higuchi's and Sevcik's methods (□, the plot needs to be zoomed to show the details) that are in the period of all 19 earthquakes. The fact that there are three techniques that show this behaviour makes these fractal disturbances noteworthy. However, it is not possible to discriminate whether these disturbances are post-seismic activity of the great Wenchuan earthquake or pre-seismic activity of another one in the area, and this is a limitation of the present methodology. On the other hand, observing the data of Table  2 carefully, it can be seen that earthquakes 13–18 are practically the seismic sequence of the Wenchuan earthquake since all occurred on the same day as earthquake 19. Especially 16–18 happened at the same hour. Moreover, earthquakes 11 and 12 are also within the post sequence of the Wenchuan earthquake because they occurred just one (11) and two (12) days after. Moreover, all earthquakes from 8 to 19 occurred in the same month (April 2008). These peculiar facts complicate discriminating the above concurrent findings using the three techniques as post- or pre-seismic. This is not the case for the three co-occurrences of the combinations of the fractal dimension methods (*pentagon*—yellow, namely, Sevcik's versus Katz's, and Higuchi's methods (three techniques); *pentagon*—blue, viz. Katz's versus Higuchi's and Sevcik's methods (three techniques); and green ◁, namely, Higuchi's versus Katz's and Sevcik's methods (three techniques)). These occurred prior to events 7, 6, and 5 but could be prior to 4, 3, 2, and 1 as well (decreasing time distance between fractal disturbance and earthquake occurrence) (all these sub-figures need to be zoomed as well).

The most significant finding of this paper is left for the last section. It can be clearly observed in all sub-figures that the vast majority of coincidences are prior to the great Wenchuan earthquake (19) and its synchronous post-earthquakes (10–19). This is the most important observation that the reader should emphasise. It is not only one method but all 13 methods that coincide. Moreover, during this preparation phase of the great Wenchuan earthquake, there are far more coincidences for even three or fewer techniques as well (such

as red ▷ in sub-figure b, namely, power-law analysis versus Katz's and Sevcik's methods (three techniques), and yellow ⊡, that is, DFA versus Higuchi's and Sevcik's methods (three techniques)). In the most emphatic manner, it is here declared that all these fractal disturbances (importantly, with results from the meta-analysis) are, most possibly, due to the great Wenchuan earthquake. This is the most important finding of this paper, which we leave for last to emphasise. It is the great magnitude of the Wenchuan earthquake, its post-seismic activity, and the novel combining–linking of several different fractal techniques that have managed to allow such an important finding to be outlined.

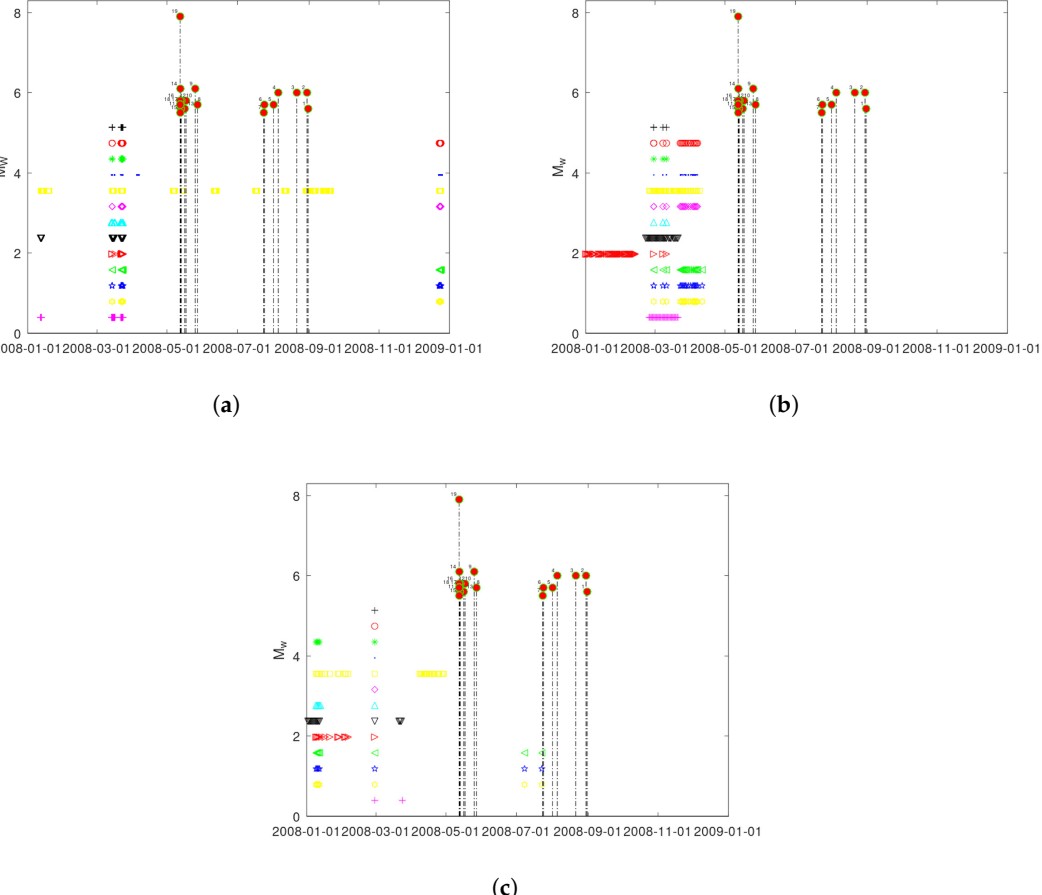

**Figure 18.** Overview of the full computational meta-analysis results by all thirteen selected combinations of fractal methods per five, four, three, and two methods. Data from (**a**) KDS, (**b**) GS, and (**c**) MSS. Symbols: "+" (black): DFA versus all methods (5 techniques—DFA, Power law analysis, Katz's, Higuchi's, and Sevcik's methods); "⊙" (red): DFA versus all fractal dimension techniques (4 techniques); "∗" (green): power-law analysis versus all fractal dimension techniques (4 techniques); "." (blue): DFA versus Higuchi's and Katz's methods (3 techniques); "⊡" (yellow): DFA versus Higuchi's and Sevcik's methods (3 techniques); "◇" (magenta): DFA versus Katz's and Sevcik's methods (3 techniques); "▽" (cyan): power-law analysis versus Higuchi's and Katz's methods (3 techniques); "▽" (black): power-law analysis versus Higuchi's and Sevcik's methods (3 techniques); "▷" (red): power-law analysis versus Katz's and Sevcik's methods (3 techniques); "◁" (green): Higuchi's versus Katz's and Sevcik's methods (3 techniques); "⌂" (yellow): Sevcik's versus Katz's and Higuchi's methods (3 techniques); "⌂" (blue): Katz's versus Higuchi's and Sevcik's methods (3 techniques); "+ ' (magenta): DFA versus power-law analysis (2 techniques). Horizontal axis is in actual dates.

As final remarks of this paper, it should be noted that the overall approach of combining different fractal techniques by employing thresholds on the final ASCII outcomes is a significant novelty of this paper that should be emphasised. Moreover, according to the

references of this work, there are only a few papers that report different methods in evidencing possible anomalies associated, most probably, with seismic activity. This is also a significant novelty. In addition, this approach has also worked well with air pollution $PM_{10}$ time series [37,39,61]. This latter fact makes the methodology and interpretations useful for other scientific areas as well and is very promising for the continuation of this work.

## 5. Conclusions

This study investigated the fractal patterns hidden in one-year radon in groundwater disturbances derived from five stations in China before and after the devastating Wenchuan ($M_w$ = 7.9) shallow (depth = 19 km) earthquake that occurred on 12 May 2008 (*day* 133). The data were analysed with five distinct fractal techniques (DFA; fractal dimensions with Higuchi's, Katz's, and Sevcik's methods; and power-law analysis). Sliding windows of step 1 were utilised in segmented portions glided throughout each signal. Via literature-based thresholds, several notable areas were found in the fractal variations of the KDS, GS, and MSS data, whilst non-significant fractal portions were found in the signals of PZHS and SPS. Up to *day* 133 (12 May 2008), several critical epochs were located in the signals of KDS, GS, and MSS. Specifically, the DFA exponents during these epochs were $1.5 \leq \alpha < 2.0$ ($0.5 \leq H < 1.0$), whereas several exponents were above 1.8 ($0.8 \leq H < 1.0$). The fractal dimension epochs exhibited Katz's fractal dimensions around 1.0 and 1.2 ($0.8 < H < 1$), Higuchi's dimensions between 1.5 and 2 ($0 < H < 0.5$), and Sevcik's dimensions between 1 and 1.5 ($0.5 < H < 1.0$). Several power-law $b$ exponents were above 1.7, and numerous were above 2.0. All these are in agreement with precursory fractional Brownian motion signal parts. The differentiations between the KDS-GS-MSS and the PZHS-SPS ones could be attributed to the geological background with the theories of asperities (fractional Brownian motion profile relative roughness) and selectivity. As a systematic last action, all results of KDS-GS-MSS were analysed using a novel, two-step, fully computerized methodology that located the exact out-of-threshold fractal areas and combined the outcomes of the different methods per 5, 4, 3, and 2 (maximum 13 common combinations) in association with the 19 earthquakes of $M_w \geq 5.5$ in the greater area. The vast majority of the different combinations of techniques showed coincidences prior to the great Wenchuan earthquake and its synchronous post-earthquakes. This important finding was justified not only with one method but also, in many cases, with all 13 different methods. Critical epochs after the Wenchuan earthquake could be attributed to other earthquakes in the area, whereas a post-seismic view can also be accepted. The combined results, in association with the great earthquake's magnitude and small depth, make the findings of this paper promising for earthquake-related studies.

**Author Contributions:** Conceptualisation, D.N.; data curation, A.A. and D.N.; formal analysis, A.A., D.N. and N.W.; investigation, A.A., D.N. and N.W.; methodology, A.A. and D.N.; resources, A.A. and N.W.; software, D.N.; supervision, D.N. and N.W.; visualisation, A.A. and D.N.; writing—original draft, D.N.; writing—review and editing, A.A. and N.W. All authors have read and agreed to the published version of the manuscript.

**Funding:** This research received no external funding.

**Data Availability Statement:** Not applicable.

**Conflicts of Interest:** The authors declare no conflict of interest.

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
