# Peer review of "Fractal Patterns in Groundwater Radon Disturbances Prior to the Great 7.9 Mw Wenchuan Earthquake, China"

_geosciences, doi:10.3390/geosciences13090268_

Round 1
Reviewer 1 Report
The paper "FRACTAL PATTERNS IN GROUNDWATER RADON DISTURBANCES PRIOR TO THE GREAT Mw=7.9 WENCHUAN EARTHQUAKE, CHINA" explores the fractal patterns concealed in one-year radon in groundwater disturbances obtained from five locations in China before and after the disastrous Wenchuan earthquake, by suing DFA techniques. This topic is attractive to seismologists and physicists. The manuscript is overall well written, but the introduction should be improved. Besides, some typographical errors must be corrected, for example, in line 24 where the full stop after citations [2–9] is missing. Also, Line 43: Please remove one of the closing parentheses in "(Figure 1))". Please see the comments below.
(1) In lines 26-27, the authors write: "Earthquakes are inherently complex, thus, several techniques and multi-level strategies are required [7].". Various techniques and approaches are needed for what purpose? For earthquake prediction or precursors? For locating hypocenters? The authors should clarify this stating.
(2) Figure 1 is mentioned in the body text only in the sense of presenting the Wenchuan earthquake. The authors must accurately describe this figure in the manuscript by associating the epicenter with the radon in groundwater monitoring station distributions and the geological setting.
(3) In the Introduction section, the data description (when the authors refer to Figure 2) is inadequate for this section. Authors should discuss details of the data used in a data section. Instead, the authors should discuss other cases where fractal patterns were observed. Also, the motivation and contributions of this work should be better clarified.
(4) In the caption of Table 1 and some excerpts from the manuscript, the authors write "distance from epicentre". Please consider replacing it with "Epicentral distance".
Typographical errors must be corrected.
Author Response
POINT TO POINT RESPONSE TO REVIEWER R1
We would like to thank the reviewer for considering our manuscript and providing comments. We appreciate the valuable time spent.
To assist the reconsideration of our paper we provide two discrete files additional to the whole revision:
- PaperWenchuan_v10_cm.pdf: This pdf (created by pdlatex) shows the corrections & insertions as highlighted text. We followed the next rules:
a1. Yellow highlight: Changes in expressions indicated in the annotated pdf of R2
or made by us.
a2. Strikethrough text: Deleted expressions
a3. Cyan highlight: Changes according to comments of R1
a4. Other changes that can not be shown in LaTEX are given in this point-to-point response
(b) PaperWenchuan_v10.pdf This pdf (created by pdlatex) is the final clean paper
Many expression problems have been located and changed. We reworded several phrases to enhance the text further and to avoid plagiarism. All these are shown in yellow highlight. Your specific comments are shown in cyan highlight.
As a significant problem found in captions by us, wherever the year 2018 was written it was corrected to 2008!
We hope that we responded adequately
SPECIFIC COMMENTS
Comment 1:
The paper "FRACTAL PATTERNS IN GROUNDWATER RADON DISTURBANCES PRIOR TO THE GREAT Mw=7.9 WENCHUAN EARTHQUAKE, CHINA" explores the fractal patterns concealed in one-year radon in groundwater disturbances obtained from five locations in China before and after the disastrous Wenchuan earthquake, by suing DFA techniques. This topic is attractive to seismologists and physicists. The manuscript is overall well written, but the introduction should be improved. Besides, some typographical errors must be corrected, for example, in line 24 where the full stop after citations [2–9] is missing. Also, Line 43: Please remove one of the closing parentheses in "(Figure 1))". Please see the comments below.
Response:
We thank the reviewer for the kind comments.
We reworded several phrases sand many awkward phrases and typographical errors (among other issues). We found also other problems and tried to solve them all.
Kindly refer to the yellow highlighted text for expression changes and in cyan highlighted text for your specific comments.
We hope that we implemented this phase correctly.
Comment 2:
In lines 26-27, the authors write: "Earthquakes are inherently complex, thus, several techniques and multi-level strategies are required [7].". Various techniques and approaches are needed for what purpose? For earthquake prediction or precursors? For locating hypocenters? The authors should clarify this stating.
Response:
Kindly accept our reply here.
We believe that earthquakes are complex phenomena. They are described by fractals, critical points, roughness measures, complex procedures and self organisation. We have the aspect that various techniques are needed so as to provide more evidence to evaluate if some kind of anomaly (fractal, entropy and so on) might be due to a geo-physical procedure that generated an earthquake.
Later in the Introduction, the methods the results we state this aspect with different expressions. Reference 7 is indicative. In our reply to comment 7 of R2’s annotated file, we provide some more references. Related papers to the aspect of various techniques are the following
@Article{ eftaxias-etal-2008,
author = {K. Eftaxias and Y. Contoyiannis and G. Balasis and K.
Karamanos and J. Kopanas and G. Antonopoulos and G.
Koulouras and C. Nomicos},
title = {Evidence of fractional-Brownian-motion-type asperity model
for earthquake generation in candidate pre-seismic
electromagnetic emissions},
journal = {{N}at. {H}azard {E}arth {S}ys.},
year = {2008},
volume = {8},
number = {},
pages = {657-669}
@Article{ eftaxias-etal-2009,
author = {K. Eftaxias and G. Balasis and Y. Contoyiannis and C.
Papadimitriou and M. Kalimeri and L. Athanasopoulou and S.
Nikolopoulos and J. Kopanas and G. Antonopoulos and C.
Nomicos},
title = {Unfolding the procedure of characterizing recorded ultra
low frequency, kHZ and MHz electromagnetic anomalies prior
to the L\textquoteright Aquila earthquake as pre-seismic
ones-Part 1},
journal = {{N}at. {H}azard {E}arth {S}ys.},
year = {2009},
volume = {9},
number = {},
pages = {1953-1971}
}
@Article{ eftaxias-etal-2010,
author = {Eftaxias, K. and Balasis, G. and Contoyiannis, Y. and
Papadimitriou, C. and Kalimeri, M.},
journal = {NHESS},
number = {2},
pages = {275-294},
title = {Unfolding the procedure of characterizing recorded ultra
low frequency, kHZ and MHz electromagnetic anomalies prior
to the L'Aquila earthquake as pre-seismic ones - Part 2},
volume = {10},
year = {2010}
We discussed this issue a lot in the discussion section.
We hope that our reply is adequate. We can make further alterations in a second revision if needed.
Comment 3:
Figure 1 is mentioned in the body text only in the sense of presenting the Wenchuan earthquake. The authors must accurately describe this figure in the manuscript by associating the epicenter with the radon in groundwater monitoring station distributions and the geological setting.
Response:
We think that we used this figure in our discussion in the manner indicated in your comment. Kindly refer to lines 423-466 (initial manuscript) (paragraph, “The differentiation of the KDS. GS, MSS stations….”). Not only we make use of this figure but we provide possible explanation why PZHS and SPS do not give indications.
If there is something more to be done, we will be glad to implement it.
Comment 4:
In the Introduction section, the data description (when the authors refer to Figure 2) is inadequate for this section. Authors should discuss details of the data used in a data section. Instead, the authors should discuss other cases where fractal patterns were observed. Also, the motivation and contributions of this work should be better clarified.
Response:
Please accept our apologises. When writing the LaTEX paper, by mistake we used ~\ref{Fig-2} in the Introduction. We corrected this to the correct ~\ref{Fig-1}. So this inadequacy was corrected. Data are discussed in data sections.
Regarding other cases where fractal patters are observed, we have expressed many related cases (with references) in section 3.1 (paragraph 1), section 3.2, section 3.3( paragraph 1), section 3.5 (paragraph 1).
If you think that this information should be better if they are moved somewhere before, we can do it in a second revision. If additional information should be given, we can also do it in a second revision. Obviously, since we submitted in thisway, we prefer the presentation organised per method. But, honestly, this is not strict.
Regarding motivation, we think that we put it in paragraph 2 (Earthquakes are inherently complex, thus, several techniques and multi-level strategies are required) and paragraph 3 (One of the key components of the present research..) of the Introduction. What we mean is that since earthquakes are complex, and the specific earthquake is very large, it is worth to study.
However, our way of thinking and expressing is constructive, (you may have noticed that in the paper already. We provide initial information, more aspects, different viewpoints up to what we think is important).
In this constructive manner we have expressed the motivation: Fractal patterns in Radon, ULF, HF, VHF electromagnetic data, remote sensing etc. Pre-Seismic precursors in general. Then a very big earthquake of Mw= 7.9, shallow. Many lives lost. Properties damaged. We believe this is a good motivation to make a study. We reworded the expression “One of the key components” to “The motivation” although we think that this change limits the expression.
Of course the term motivation can be used differently and present only the most significant parts. If you think that this is the best way, we can do so in the second revision.
The contribution is written in many parts of the results and of course in the conclusions.
Again we can employ different ways of expressing.
We did some rewording but we can do more in the second revision depending on your views.
Comment 5:
In the caption of Table 1 and some excerpts from the manuscript, the authors write "distance from epicentre". Please consider replacing it with "Epicentral distance”.
Response:
We thank the reviewer for this comment. Wherever this could be done we implemented in text as well. Please kindly refer to the yellow highlights
Comments on the Quality of English Language
Typographical errors must be corrected.
Response:
We did significant work on that. If something can be altered further we will be glad to do it.
Reviewer 2 Report
Dear authors, this is an interesting paper. The methodological approach appears sound. (I think that there are some errors in formulas.) For details please see the annotated pdf manuscript, attached.
As a detail, did you check what the effect of choosing a different segmentation or window size would be? Another issue is uncertainty. The fractal measures are results of statistical procedures applied to data and therefore have by nature uncertainty. I know that these are probably difficult to estimate; nevertheless they could be helpful for interpretation.
A further point: I think that you should address the question, how the technique could be applied to prediction in practice, when one has only a series of measurements up to a point *before* a possible earthquake. This study is a diagnostic one, performed after the quake has happened, so to speak a postdiction
However, the ms. is very difficult to read due to poor English. In my opinion, the results section is not well structured and very difficult to follow. Above all this applies to the interpretation of the graphs: this is of course important, but really tedious to read in its current form. Please think about re-structuring. Perhaps you can generate a "synoptic" table in which the main results are concisely summarized: the stations would be represented by rows and the fractal methods by columns. (But this is only an idea.)

English is such that the text is sometimes difficult to read. In fact I am not sure whether I understood everything... perhaps not.
I strongly recommend editing by a proficient English writer! In its current form the text is not acceptable.
In some cases, I tried to suggest some improvement, but nevertheless the text needs a linguistic overhaul.
I think that assessment of the ms. will be easier once this has been done!
But as said, overall an interesting paper.
Author Response
REBUTTAL and POINT TO POINT RESPONSE TO REVIEWER R2
We would like to thank reviewer R2 for considering our manuscript and providing comments. We appreciate the valuable time spent.
To assist the reconsideration of our paper we provide two discrete files additional to the whole revision:
- PaperWenchuan_v10_cm.pdf: This pdf (created by pdlatex) shows the corrections & insertions as highlighted text. We followed the next rules:
a1. Yellow highlight: Changes in expressions indicated in the annotated pdf of R2
or made by us.
a2. Strikethrough text: Deleted expressions
a3. Other changes that can not be shown in LaTEX are given in this point-to-point response
(b) PaperWenchuan_v10.pdf This pdf (created by pdlatex) is the final clean paper
Below is our response to the comments. First we respond to the suggested changes in the annotated pdf of reviewer R2 for which we have a different opinion. All other changes have been adopted or enhanced further to reflect our ideas. We thank reviewer R2 for this annotated file. We also reworded several other phrases to enhance the text further and to avoid plagiarism. All are shown in yellow highlight
We hope that we responded adequately
EXPRESSIONS AND OTHER PROBLEMS ANNOTATED BY R2
In the annotated pdf file of reviewer R2, there were 141 annotations. Many of these suggested certain phrase changes, other located problems in expressions while some other expressed questions or comments regarding the claims of the paper. The latter are the important ones since they are additional to the main comments of R2.
We implemented the majority of suggested wording changes in the annotated pdf file of R2. In several cases we found bigger problems and, so, reworded the text differently to reflect our concepts better. All these changes are highlighted in yellow and the deletions as strikethrough text. As a significant problem found in captions by us, wherever the year 2018 was written it was corrected to 2008!
(a) For the cases for which additional comments are expressed, we reply here. (b) For the cases for which we have different aspects from the reviewer’s annotations, we explain the details of our differentiations (additionally to our reply the main comments of R2). (c) For the cases for which we strongly disagree, we provide a rebuttal in this section as well.
For assisting the review we have to remark that many of the bubble-comments were not presented in the correct places (as seen in our MACOS environment), hence for these cases the lines of our reply (in this section) are indicative.
We go now into details of cases (a), (b) and (c):
1.Line 24, annotated pdf file.
Comment:”this sentence is too general. This is true for nearly every natural phenomenon.”
REPLY: Complex processes are non-linear. They refer to roughness of profiles, to fractal behaviour, to SOC traces and so on. There are phenomena that are not complex. For example radiation transfer trough matter. There, the phenomenon is random and obeys a certain statistical law (Poisson statistics). It is a stochastic phenomenon. Not a complex one. If the processes were random Hurst exponents would be 0.5, DFA exponents 1, Power-law exponents 2 and fractal dimensions 1. In fact, the windows for which these numbers are recorded, are stochastic. The other are non-stochastic. Moreover, geo-physics before earthquakes obey complex processes at critical points.
2. Line 28, annotated pdf file
Comment: “gradual downscaling of time, location, and magnitudes [2]. please explain. Seems to be an important point.”
REPLY: This expression is similarly written in the well-known review of Ciccerone et al [2]. It is a standard approach in seismology, to try to reduce the range of values for precursory time and pre-estimated magnitude and, possibly locations. All these approaches exist in the literature for radon. Kindly see also our review on this [7,8].
3.Line 36, annotated pdf file
We reworded the whole expression in order to reflect what is written in [1]. We moved the reference earlier (can not be shown in laTex pdf).
4.Lines 49-50, annotated pdf file
We reworded the whole sentence.
5.Line 54, annotated pdf file
Comment: “unavoidable occurrence” do you want to say that such "features" lead unavoidably, i.e., necessarily to earthquakes? I don't think that this is correct.
REPLY: We have a completely different view of the subject here with reviewer R2. It is expressed in several publications, laboratory experiments and many many researches, that when the system goes into a SOC (self organised critical) state, the occurrence of the earthquake becomes unavoidable. Some example papers are given below:
@Article{ balasis-etal-2016,
author = {Balasis, G. and S. Potirakis and M. Mandea},
itle = {Investigating Dynamical Complexity of Geomagnetic Jerks Using Various Entropy Measures, Front},
year = {2016},
journal = {Earth. Sci},
number = {71},
pages = {1-10},
volume = {4},
}
@Article{ eftaxias-etal-2007b,
author = {K. Eftaxias, V. Sgrigna, T. Chelidze},
title = {Mechanical and electromagnetic phenomena accompanying
preseismic deformation: From laboratory to geophysical
scale},
journal = {{T}ectonophysics},
year = {2007},
volume = {341},
number = {},
pages = {1-5}
}
@Article{ eftaxias-etal-2008,
author = {K. Eftaxias and Y. Contoyiannis and G. Balasis and K.
Karamanos and J. Kopanas and G. Antonopoulos and G.
Koulouras and C. Nomicos},
title = {Evidence of fractional-Brownian-motion-type asperity model
for earthquake generation in candidate pre-seismic
electromagnetic emissions},
journal = {{N}at. {H}azard {E}arth {S}ys.},
year = {2008},
volume = {8},
number = {},
pages = {657-669}
}
@Article{ ida-etal-2012,
author = {Ida,Y., Yang and D. Li and Q., Sun and H., Hayakawa and M. },
journal = {Nonlin. Processes Geophys.},
pages = {577-583},
title = {Fractal analysis of ULF electromagnetic emissions in possible association with earthquakes in China.},
volume = {19},
year = {2012},
}
@Article{ contoyiannis-etal-2004,
author = {Contoyiannis, Y. F. and Diakonos, F. K. and Kapiris, P. G.
and Peratzakis, A. S. and Eftaxias, K. A.},
journal = {Phys. Chem. Earth},
number = {4-9},
pages = {397-408},
title = {Intermittent dynamics of critical pre-seismic
electromagnetic fluctuations},
volume = {29},
year = {2004}
}
@Article{ minadakis-etal-2012,
author = {G. Minadakis and S. Potirakis and C. Nomicos and K.
Eftaxias},
title = {Linking electromagnetic precursors with earthquake
dynamics: An approach based on nonextensive fragment and
self-affine asperity models},
journal = {{P}hysica {A}},
year = {2012},
volume = {},
number = {},
pages = {},
doi = {10.1016/j.physa.2011.11.049}
}
@Article{ skordas-2014,
author = {Skordas, E. S.},
journal = {Chaos},
pages = {023131},
title = {On the increase of the "non-uniform" scaling of the
magnetic field variations before the M(w)9.0 earthquake in
Japan in 2011},
volume = {24},
year = {2014}
}
@Article{ smirnova-etal-2004,
author = {N.A. Smirnova and M. Hayakawa and K. Gotoh},
title = {Precursory behavior of fractal characteristics of the ULF
electromagnetic fields in seismic active zones before
strong earthquakes},
journal = {{P}hys. {C}hem. {E}arth},
year = {2004},
volume = {29},
number = {},
pages = {445-451}
}
@Article{ smironva-and-hayakawa-2007,
author = {N.A. Smirnova and M. Hayakawa},
title = {Fractal characteristics of the ground-observed ULF
emissions in relation to geomagnetic and seismic
activities},
journal = {{J}. {A}tmos. {S}ol. {T}er. {P}hy.},
year = {2007},
volume = {69},
number = {},
pages = {1833-1841}
}
@Article{ telesca-etal-2004,
author = {Telesca, L. and Lapenna, V. and Macchiato, M.},
journal = {Chaos Solit. Fractals},
pages = {1-15},
title = {Mono- and multi-fractal investigation of scaling
properties in temporal patterns of seismic sequences},
volume = {19},
year = {2004}
}
6. Line 56, annotated pdf file
REPLY: We thank the reviewer for that. By mistake section 2.2 was written as \subsection and not as \subsubsection which is the correct. Now former 2.2 became 2.1.2 (automatically in LaTEX). We checked all LaTEX for such inconsistencies.
7.Line 61, annotated pdf file
Comment: “Fig. ?”
REPLY: We forgot to put the phrase Fig. Inside. Latex enumerated figure correctly. It was our omission. We corrected it.
8.Line 86, annotated pdf file
Comment: “ionization”
REPLY: Ionisation (English spelling) and scintillation are different phenomena. We think our expression is valid.
9.Line 91, annotated pdf file
Comment: “explained or described?”
REPLY: To our opinion fractals explain, delineate, clarify the phenomena that have fractal properties. There is no great differentiation, more or less. Both expressions are valid.
10. Lines 96-97, annotated pdf file
Comment: what are the constituent parts of a fractal?
REPLY: We deleted these lines. Nothing very important.
11.Line 107, annotated pdf file
Comment: “Therefore these techniques were also udes in this paper.”
REPLY: We are sorry, but we did not understand what to change and where.
12. Line 121, annotated pdf file
Comment: 1) Please give definition of H, 2) how did you calculate it?
REPLY: We are sorry, but Hurst exponent is a basic knowledge in the related literature. By the way, we provide references 30-51 which all give the inspiration, definition of Hurst exponent among other information. In the methods section we present all the equations to convert the various exponents to Hurst exponent, under the constraint of a linear one-to-one equation. We have expressed criticisms for this in our previous publication, but it a well-known approach. We provide several related references in the methods section.
13.Line 132, annotated pdf file
Comment: “power-law connections” not defined or explained! Also the term erratic should be defined more precisely: what is the criterion to call a signal "erratic"? (contrary to "regular"?) - I think that this is important in this context.
REPLY: We are sorry, but power-law behaviour is trivial in non-linear geophysics. By the way we have given full description of how we have calculated the power-law fractal analysis. Regarding the “erratic” term. Please refer to reference 52 from the Varotsos team (fist name prof. Skordas).
14. Lines 163-164 annotated pdf file
Comment: error in the formula
REPLY: It is correct. See all our related publications.
15. Lines 205-207, annotated pdf file
Comment a: 1) error?, should be sqrt((t\i-t\i+1)^2+(y\i-y\i+1)^2), 2) t and y have different units. What is the unit of dist?
REPLY: a It is correct. It is Euclidean distance! That means the metric in vertical and horizontal axis is the same. s1^2=t1^2+y1^2, s2^2=t2^2+y2^2. So it is correct. Please see references 78 and 79. Kindly, check this.
Comment: b should be N instead of n ?
REPLY b: No. It’s n! n is described next to this equation!
16. Lines 208-209, annotated pdf file
Comment: this is not a length in the same sense as in 3.4.1
REPLY: This is a length as described in our reply above (reply comment 15). Kindly check reference 80.
17. Lines 217-219, annotated pdf file
Comment: Lines L/(2 eps) why factor 2? Acc. to the def. in the last line, I would expect N=L/eps. Please check again with ref.79, section E. There, the segments have length 2eps.
REPLY: It is correct! Kindly check for example a different reference:
InternatIonal Journal of Computer Science and Technology, ISSN : 0976-8491 (Online) | ISSN : 2229-4333 (Print) JCST Vol. 4, Issue Spl - 2, AprIl - June 2013 ,Sevcik’s Fractal Based Dimensionality Reduction of Hyper-Spectral Remote Sensing Data, Nikhil Sharma, Dr. J. K. Ghosh
18.Line 279, annotated pdf file
Comment: subsection 3.5.1, same subsection
REPLY: It was a mistake in the ~\ref command. Now is fixed
19. Line 306, annotated pdf file
Comment: persistency has been defined above for H but not for DFA exponent alpha. This is addressed below, but should be mentioned here.
REPLY: This is really of negligible value because there are so many references given which support to all these claims. Persistency and anti persistency are well known tendencies in non-linear physics. If we define them for every exponent we would repeat information of non-added value. If the reviewer insists, we will do so on the second revision.
20. Line 391 and others
Comment: please reconsider the usage of the word “successful"
REPLY: We changed the term to successive
21. Line 400., annotated pdf file
REPLY: Fixed (error in latex \cite command),
22. Figure 1 and others.
Comment:it would be useful to indicate the time of the earthquake in the graphs. the x-axis of the Rn series should be the same in all figures, best in days.
REPLY: The time series graph is in correct time. The analysis is NOT in absolute time (explained in methods section) It is in window numbers (one window is omitted). Very different. It is the correct presentation. We were vey careful on that issue!!
23.Line 410, annotated pdf file
Comment: “values”
REPLY: It is of precursory value=importance, worth, usefulness (as a pre-earthquake precursor)
24.Line 420, annotated pdf file
Comment: distance?
REPLY: We used “epicentral distance”. R1 remarked that as well.
25. Lines 430-341, annotated pdf file
Comment: please rephrase
REPLY: implemented
26. Lines 432, annotated pdf file
Comment: sentce is not understandable
REPLY: We deleted the awkward phrase. We rephrased text accordingly.
27. Line 490, annotated pdf file
Comment: I am concerned about this "error" (as also before).
What is the source of the error?Could it not be an error in the method?
REPLY: Computations many, many times result in error calculations. Every programmer is aware of that. It is called bug in the code, because it is disturbing as a bug and cannot be easily corrected. In most of the cases might just be a logical error and not a coding error. This approach has been employed in all false calculations of our papers. These codes do not exist commercially. These are methods conceptualised and coded by us. They programs are continuously enhanced.
28. Line 587, annotated pdf file
Comment: “Corresponding signs are not enough” ? Do you mean something like this: fractal indicators are not sufficiently significant ?
REPLY: We are deeply concerned about the fact that things written in text are not understood. We write it in several parts, that several papers stay only to the identification of an anomaly. Many others go into the fractal and SOC trends of the time-series,
which are hidden in this time-series. The hidden traces reflect what’s going on in the crack generation. Organisation and evaluation. All these are critical epochs. We write that, because for us it not just the identification of an anomalous FRACTAL behaviour, but the parallel finding with DIFFERENT TECHNIQUES and even better from the DATA OF DIFFERENT STATIONS.
Hence, for some enough is just a radon anomaly. For others, just a critical fractal finding. For us, what we write above, in the paper and all the related discussion.
29. Line 595, annotated pdf file
Comment: “This multi figure conceals all the important findings.” ?? If it conceals the important findings, what is the figure good for?
REPLY: We think that the reviewer did not understand what the multifigure present. We believe that we have explained this very well in text, we provided reasoning for our methods (combination of techniques) and why these are important. We have nothing else to clarify more here.
30. Line 611, annotated pdf file
Comment: “Special software” which?
REPLY: Please kindly refer to methods and the reply to comment 27 above. There exist NO COMMERCIAL SOFTWARE for that. It is our conceptualisation, our development, our software writing, debugging and so on. That means special software.
31. The suggestions in abstract were not implemented because it precisely in the limit of 200 words. If we set the Abbreviations in Abstract we would not be able to present results Regarding the comment for the maximum of 13 combination of methods it is well expressed. Regarding the term critical epochs, we have responded already and we respond further in our response to the mani core of comments of R2.
SPECIFIC COMMENTS
Comment 1:
Dear authors, this is an interesting paper. The methodological approach appears sound. (I think that there are some errors in formulas.) For details please see the annotated pdf manuscript, attached.
Response:
We thank the reviewer for the kind words on our paper. Since in the annotated pdf file there were 141 comments, there was a lot of work. In fact we were impressed by your detailed review and work implemented by the reviewer. Many thanks. We adopted the majority of the expression suggestions and where we found a bigger problem we solved that.
As you can see in our selected 30 responses to your 141 annotated comments there are cases where we disagree. This refers to errors in formulas. Kindly refer to section (EXPRESSIONS AND OTHER PROBLEMS ANNOTATED BY R2) (uploaded to your response as well, but included in the cover letter) and specifically (for the errors) to our reply to your comments 15,16,17. We still believe that the formulas are correct
Comment 2:
As a detail, did you check what the effect of choosing a different segmentation or window size would be?
Response:
We thank the reviewer for this comment. Not only we have checked the results of one different segmentation but many. In all kinds of signals we check to find the best segmentation. In our publications we have employed different windows. For us this is a trivial procedure. We have analysed so-far so many signals and with so-many techniques that we can not even describe it.
Comment 3:
Another issue is uncertainty. The fractal measures are results of statistical procedures applied to data and therefore have by nature uncertainty. I know that these are probably difficult to estimate; nevertheless they could be helpful for interpretation.
Response:
Kindly allow us here to repeat some information that we expressed in our partial responses.
Statistical procedures and fractal measures are completely different concepts. Statistical procedures describe stochastic phenomena (example: radiation transfer trough matter. Obeys a certain distribution, Poisson statistics). Fractal measures obey power-laws. That is non-linear processes. Fractal measures are strongly related to complex patterns (complexity), SOC traces, rough profiles, self-organisation. In fact there are stochastic and non-linear processes in nature. Stochastic many times can be deterministic. There are several Monte Carlo codes which estimate mean values, uncertainties (second moment), Kurtosis and many other moments and statistical measures. In fact in stochastic phenomena statistics and Markov chains play a significant role. They might be called and deterministic.
On the other hand, fractal, critical phenomena, fractional movements are chaotic. They might go into definite solutions within a bifurcation. These definite solutions result in the unavoidable evolution when reached. A butterfly sits on the roof and the roof collapses. We wash glasses and one glass is hit and suddenly breaks into pieces (can not be avoided). These phenomena are non-Markovian (we write all the details in the paper)
These are some examples that justify that justify why your comment can not be replied. By nature they do not have the uncertainty, in the way implied.
Moreover, geo-physics before earthquakes obey complex processes at critical points.
Comment 4:
A further point: I think that you should address the question, how the technique could be applied to prediction in practice, when one has only a series of measurements up to a point *before* a possible earthquake. This study is a diagnostic one, performed after the quake has happened, so to speak a postdiction
Response:
Kindly, the term “post-diction” does not exist in papers.
Second, we do not use the term prediction because we do not believe in that. And this because all analysis is done after the events, with one target of the earthquake scientific community: To progress the research and find credible precursors that can be used quite well in a phase before earthquake. This requires collaborations and dedicated funds, we believe.
Third, our way of thinking and expressing is constructive, (you may have identified in the paper already. We provide initial information, more aspects, different viewpoints up to what we think is important). This is how we put it in the paper. We provide initial evidence with DFA (very robust). Discuss it. Then add FD analysis. Partial results. Implications. Discuss it. Power-law (gold-standard). Discuss it. Finally combine all methods and from different stations. Computationally.
In our way of thinking and writing (as you might have found) we employ the phrases: “this is probably due to”, “might possible be pre-earthquake sign”, it could be a post-seismic sign” and similar terms. We avoid the use of prediction. We also discussed the limitations of our approach. How can we apologise for something we do not present it this way?
Wherever the term prediction is mentioned is because a related reference utilises this word (for example first paragraph, reference Conti et al 2021, second paragraph introduction, reference Cicerone et al, 2009).
Comment 5:
However, the ms. is very difficult to read due to poor English. In my opinion, the results section is not well structured and very difficult to follow. Above all this applies to the interpretation of the graphs: this is of course important, but really tedious to read in its current form.
Perhaps you can generate a "synoptic" table in which the main results are concisely summarized: the stations would be represented by rows and the fractal methods by columns. (But this is only an idea.)
Response:
Dear Reviewer, we considered your aspects so deeply that we, initially, did not intend to revise the paper. At first reviewer R1 commented that the paper is well written. The same aspect was initially expressed by the journal since we were invited after our submission to put the research in a preprint.
Your hard work on expression suggestions was more than highly appreciated.
But, Dear Reviewer, you expressed interpretations in your annotated pdf file that gave us the impression that it was not the English that were poor (several awkward phrases and typos were corrected though), but the substance of the paper. We have mentioned and discussed in our responses, disagreements in the physical principles. Not the interpretations. It is for this reasons that we wrote all these methods, presented all these small details and given so many references. You requested issues already solved and mentioned in references.
We tried hard to provide in this reply much information and try to delineate the subject. We are already in 16 pages of the total reply and 12 pages to your personal reply. We really can not do more. The synoptic table would not add something not given in text. We always avoid duplications of data in Tables and text as you know.
We reworded the manuscript according to your hard expression suggestions and was enhanced. We really thank you for that.
Comments on the Quality of English Language
English is such that the text is sometimes difficult to read. In fact I am not sure whether I understood everything... perhaps not.
I strongly recommend editing by a proficient English writer! In its current form the text is not acceptable.
In some cases, I tried to suggest some improvement, but nevertheless the text needs a linguistic overhaul.
I think that assessment of the ms. will be easier once this has been done!
But as said, overall an interesting paper.
Reply:
We implemented as much as we could and enhanced further. Again may thanks.
Reviewer 3 Report
I have found very interesting the manuscript geosciences-2486439 FRACTAL PATTERNS IN GROUNDWATER RADON DISTURBANCES PRIOR TO THE GREAT Mw=7.9 WENCHUAN EARTHQUAKE, CHINA submitted by Aftab Alam , Dimitrios Nikolopoulos and Nanping Wang. Authors presents an examination of fractals in the one-year radon levels in groundwaters observed at five locations in China in the period of the Wenchuan earthquake (Mw=7.9) that occurred on May 12, 2008. The analysis incorporates five distinct techniques, namely the use of DFA, fractal dimensions employing Higuchi, Katz, and Sevcik methods, as well as power-law analysis. The KDS, GS, and MSS station data delineate significant regions of fractal analysis. Fractal dimensions are observed to display Katz's D within the range of 1.0-1.2, Higuchi's D within the range of 1.5-2.0, and Sevcik's D within the range of 1.0-1.5. A significant number of power-law exponents exceed 1.7. The fractal results obtained from the KDS-GS-MSS stations are subjected to a new computerised methodology for further analysis. This methodology effectively identifies the precise regions of fractals that fall beyond the established threshold. Additionally, it integrates the findings from various approaches for comparing the occurrences of earthquakes with magnitudes of five, four, three, and two, against nineteen earthquakes with a magnitude of 5.5 in the wider geographical area. I think that only a few papers report different methods in evidencing possible anomalies. Authors should stress this novelty.The majority of different technique coincidences occurred before to the occurrence of the significant Wenchuan earthquake and its subsequent aftershocks. This phenomenon is not limited to a certain methodology, but rather encompasses a diverse range of 13 distinct methodologies. A significant number of distinct combinations of techniques shown correlations before to the occurrence of the Wenchuan earthquake and its subsequent simultaneous post-earthquakes. The occurrence of significant periods also following the Wenchuan earthquake can be ascribed to subsequent seismic events in the region, while the perspective of post-seismic activity is also a valid interpretation. I think that the findings of this research hold promise for earthquake-related studies due to the combined results pertaining to the magnitude and shallow depth of the significant earthquake. I hope the paper will be soon accepted and published after some minor possible improvements. In particular I find useful to add some sentences about the origin of recorded signals, which look consistent with a crustal deformative process and/or to a stress diffusion process at depth able to squeeze deep seated geofluids toward the surface
Author Response
POINT TO POINT RESPONSE TO REVIEWER R3
We would like to thank the reviewer for reading and commenting our manuscript. We appreciate the valuable time spent.
We were amazed by the deep level of understanding of our work. The evaluation of our work was very encouraging for us. We incorporated the comments raised.
To assist the reconsideration of our paper we provide two discrete files additional to the whole revision:
- PaperWenchuan_v11_cm.pdf: This pdf (created by pdlatex) shows the corrections & insertions as highlighted text. We followed the next rules:
a1. Green highlight: Changes in expressions as requested by R3
a2. Strikethrough text: Deleted expressions
(b) PaperWenchuan_v11.pdf This pdf (created by pdlatex) is the final clean paper
Below is our response to the comments. This document is part of the cover letter and also uploaded separately for convenience.
We hope that we responded adequately
Reviewer’s comments
GROUNDWATER RADON DISTURBANCES PRIOR TO THE GREAT Mw=7.9 WENCHUAN EARTHQUAKE, CHINA submitted by Aftab Alam , Dimitrios Nikolopoulos and Nanping Wang. Authors presents an examination of fractals in the one-year radon levels in groundwaters observed at five locations in China in the period of the Wenchuan earthquake (Mw=7.9) that occurred on May 12, 2008. The analysis incorporates five distinct techniques, namely the use of DFA, fractal dimensions employing Higuchi, Katz, and Sevcik methods, as well as power-law analysis. The KDS, GS, and MSS station data delineate significant regions of fractal analysis. Fractal dimensions are observed to display Katz's D within the range of 1.0-1.2, Higuchi's D within the range of 1.5-2.0, and Sevcik's D within the range of 1.0-1.5. A significant number of power-law exponents exceed 1.7. The fractal results obtained from the KDS-GS-MSS stations are subjected to a new computerised methodology for further analysis. This methodology effectively identifies the precise regions of fractals that fall beyond the established threshold. Additionally, it integrates the findings from various approaches for comparing the occurrences of earthquakes with magnitudes of five, four, three, and two, against nineteen earthquakes with a magnitude of 5.5 in the wider geographical area. I think that only a few papers report different methods in evidencing possible anomalies. Authors should stress this novelty.The majority of different technique coincidences occurred before to the occurrence of the significant Wenchuan earthquake and its subsequent aftershocks. This phenomenon is not limited to a certain methodology, but rather encompasses a diverse range of 13 distinct methodologies. A significant number of distinct combinations of techniques shown correlations before to the occurrence of the Wenchuan earthquake and its subsequent simultaneous post-earthquakes. The occurrence of significant periods also following the Wenchuan earthquake can be ascribed to subsequent seismic events in the region, while the perspective of post-seismic activity is also a valid interpretation. I think that the findings of this research hold promise for earthquake-related studies due to the combined results pertaining to the magnitude and shallow depth of the significant earthquake. I hope the paper will be soon accepted and published after some minor possible improvements. In particular I find useful to add some sentences about the origin of recorded signals, which look consistent with a crustal deformative process and/or to a stress diffusion process at depth able to squeeze deep seated geofluids toward the surface
Authors’ reply
Once again we would like to thank the reviewer for the kind comments.
Regarding the first suggestion: “I think that only a few papers report different methods in evidencing possible anomalies. Authors should stress this novelty.”
We added a whole new paragraph after the former last (now this is the last one) (new lines 652-660). In this paragraph we outline this expression but also commenting the significance of this approach to other subject areas. We have employed similar methodological parts in the added publications in this paragraph (cannot be highlighted due to LaTEX issues). References 32 and 34 (new BibTEX numbering now) were not included in the submission or the first revision to avoid self-citations. But in this new paragraph these two references are in-line with the last two sentences. For this reason we decided to include them.
Regarding the second suggestion:”I think that only a few papers report different methods in evidencing possible anomalies. Authors should stress this novelty.”
We added a whole new paragraph in section “2.1.2. Measurement setup” (new lines 90-93). In these lines we added two more references (new Latex numbering 22 and 23). In this paragraph we incorporated the expression suggested by the reviewer. The included references justify this expression and for this reason we adopted it, In fact it was an excellent expression. We considered that it fitted best the new position because elsewhere the reader would be confused since the focus later is on fractal methods.
We really thank you!
Round 2
Reviewer 2 Report
Dear authors, the ms. has been significantly improved; nevertheless some issues remain. Please see the detailed reply to your response, <review_2-230802.doc>.

much improved!
Author Response
Dear Editors and Reviewer,
We appreciate the time spent to review this paper. We have read the second report of R2 to our paper. Reviewer R2 sets a series of new issues. The reviewer's report refers again to the parts of our 14-page reply to the comments raised. Unfortunately our replies/rebuttal were considered insufficient by the reviewer. R2 selects to re-raise some of the comments. The reviewer R2 concluded that there will be no extra time for a third review due to time restrictions. For us, we believe that we have given all the answers that correspond to what we write in the paper and what we believe for our paper. In fact we have also a long experience and papers in the field as R2 stated for the reviewer's research, in the start of the second report. That means that we all might have some different views and aspects of the same subject. This is logical in science, but, to our opinion, it may not interfere in a review process with long explanations and extensive argumentation. To our view, there are significant controversies that can not be argued further. Since our reply was the best for us in references to our views and what is written in our paper, we may kindly ask the Editorial office to either 1) Sent the paper to other Academic editors to evaluate our work, our reply and the new issues raised. For the reasons stated above we will not reply to this reviewer, again. Our reply will not be seen, finally. We do not want any further rebuttal. or/and 2) Sent the paper to new reviewers or, otherwise 3) Withdraw the paper. We are very tired from this, at least, peculiar evolution. With Kind Regards